# Survival and growth of saprotrophic and mycorrhizal fungi in recalcitrant amine, amide and ammonium containing media

Åke Stenholm[1,2©], Anders Backlund[3©], Sara Holmström[1©], Maria Backlund[4‡], Mikael Hedeland[2‡], Petra Fransson[5©]*

1 Cytiva, Uppsala, Sweden, 2 Analytical Pharmaceutical Chemistry, Department of Medicinal Chemistry, Uppsala University, Uppsala, Sweden, 3 Pharmacognosy, Department of Pharmaceutical Biosciences, Uppsala University, Uppsala, Sweden, 4 SLU Artdatabanken, Swedish University of Agricultural Sciences, Uppsala, Sweden, 5 Uppsala BioCenter, Department of Forest Mycology and Plant Pathology, Swedish University of Agricultural Sciences, Uppsala, Sweden

© These authors contributed equally to this work.
‡ These authors also contributed equally to this work.
* petra.fransson@slu.se

**Data Availability Statement:** All relevant data are within the manuscript and its Supporting Information files.

## Abstract

The elimination of hazardous compounds in chemical wastes can be a complex and technically demanding task. In the search for environmental-friendly technologies, fungal mediated remediation and removal procedures are of concern. In this study, we investigated whether there are fungal species that can survive and grow on solely amine-containing compounds. One compound containing a primary amine group; 2-diethylaminoethanol, one compound with a primary amide group; 2,6-dichlorobenzamide (BAM), and a third compound containing a quaternary ammonium group; $N_3$-trimethyl(2-oxiranyl)methanaminium chloride, were selected. The choice of these compounds was motivated by their excessive use in large scale manufacturing of protein separation media (2-diethylaminoethanol and the quaternary amine). 2,6-dichlorobenzamide, the degradation product of the herbicide 2,6-dichlorobenzonitrile (dichlobenil), was chosen since it is an extremely recalcitrant compound. Utilising part of the large fungal diversity in Northern European forests, a screening study using 48 fungal isolates from 42 fungal species, including saprotrophic and mycorrhizal fungi, was performed to test for growth responses to the chosen compounds. The ericoid (ERM) mycorrhizal fungus *Rhizoscyphus ericae* showed the best overall growth on 2-diethylaminoethanol and BAM in the 1–20 g L$^{-1}$ concentration range, with a 35-fold and 4.5-fold increase in biomass, respectively. For $N_3$-trimethyl(2-oxiranyl)methanaminium chloride, the peak growth occurred at 1 g L$^{-1}$. In a second experiment, including three of the most promising fungi (*Laccaria laccata*, *Hygrophorus camarophyllus* and *Rhizoscyphus ericae*) from the screening experiment, a simulated process water containing 1.9% (w/v) 2-diethylaminoethanol and 0.8% (w/v) $N_3$-trimethyl(2-oxiranyl)methanaminium chloride was used. *Laccaria laccata* showed the best biomass increase (380%) relative to a control, while the accumulation for *Rhizoscyphus ericae* and *Hygrophorus camarophyllus* were 292% and 136% respectively, indicating that mycorrhizal fungi can use amine- and amide-containing

**Funding:** The author(s) received no specific funding for this work. The funder Cytiva provided support in the form of salaries for author ÅS, but did not have any additional role in the study design, data collection and analysis, decision to publish, or preparation of the manuscript. The specific roles of these authors are articulated in the 'author contributions' section.

**Competing interests:** ÅS affiliation to Cytiva does not alter our adherence to PLOS ONE policies on sharing data and materials.

substrates as nutrients. These results show the potential of certain fungal species to be used in alternative green wastewater treatment procedures.

## Introduction

The treatment and destruction of hazardous chemical wastes, such as process water from chemical industries, is a high-cost business connected with environmental risks and considerable energy consumption. This is especially true when desiccation followed by high temperature combustion is used. In the search for more environmental-friendly technologies, fungal mediated remediation and removal procedures are of interest. Bioremediation and biodegradation using for example fungi, bacteria, algae, or plants have developed alongside the commonly used physiochemical technologies [1] and today play an important role in both natural and engineered systems [2]. Fungi utilize bio-synthesized compounds that they employ powerful enzyme systems to depolymerize and catabolize, and in this capacity they also become of interest from the perspective of possibly catabolizing hazardous chemical compounds and transforming them to biomass.

Fungi are of fundamental importance to all ecosystems in terms of elemental cycling, and evolution of primary lifestyles (saprotrophic and symbiotic fungi) has occurred repeatedly via loss or reduction of genes for groups of enzymes [3, 4]. Saprotrophic fungi primarily facilitate organic matter decomposition, utilizing carbon (C) and nutrients from leaf litter and wood for growth [5]. Wood decomposing fungi can be further divided; fungi with the ability to selectively or simultaneously degrade persistent lignin using highly specialized class II peroxidases (white rot fungi), and fungi not able to degrade lignin which instead selectively degrade cellulose (brown rot fungi; use of Fenton chemistry) [3]. Ectomycorrhizal fungi (ECM), on the other hand, live in symbiosis with vascular plants and exchange mineral nutrients and water in return for photoassimilated C [6]. Depolymerization of organic matter was earlier assumed to be carried out only by free-living saprotrophic fungi, and although the involvement of ECM fungi in decomposition of soil organic matter remains controversial recent findings support the view that ECM fungi also have the capacity to oxidize organic matter [4, 7, 8], through enzyme systems similar to those of white rot fungi including peroxidases [9, 10] and Fenton chemistry of brown-rot fungi [11]. In addition to their ability to decompose organic matter, fungi were recently high-lighted for their large potential to be exploited further for industrial use; for example to improve waste disposal [12]. Fungi are well known to tolerate and metabolize both recalcitrant and toxic compounds, and are used for bioremediation [13, 14], and due to their diverse metabolic capacity fungi are good candidates for managing chemical waste.

Fungi with peroxidases are often used in whole cell fungal treatments (*in vivo*) of wastewaters, for example those that contain pharmaceuticals [15, 16] where the catabolism is performed by secreted extracellular enzymes [17, 18]. There are also examples of *in vitro* experiments in which solely enzymes have been used [19, 20]. It has been demonstrated that fungi which produce these extracellular enzymes can use nitrogen (N) containing aromatic compounds as sole N sources [21], or as both C and N source [22]. However, when it comes to the removal of non-aromatic compounds, the use of these fungi is relevant only when combined with redox-mediators that enhance the oxidation capacity of the enzymes [20] or with reactive oxygen species such as the hydroxyl radical [23].

Amines, amides and quaternary ammonium compounds are relatively commonly occurring substances, containing both N and C, which make them interesting from a nutrient point

of view. On the other hand, the removal of these substances as contaminants in ground and wastewaters is important since many of them are both toxic and carcinogenic. Amines are used in the syntheses of azo-dyes, polyurethane, pesticides and many other products. The degradation of amines is facilitated both by Advanced Oxidation Procedures (AOPs) and non-AOPs such as biodegradation [24]. AOPs are based on the generation of hydroxyl radicals which can be facilitated chemically (Fenton´s reagent), photo chemically (UV/$TiO_2$/$H_2O_2$, $O_3$,/UV) or sonolytically (ultrasound). Although argued that these techniques have an advantage of removing even the non-biodegradable contaminants, there are some drawbacks; reaction products can be even more toxic than the precursors [25], and the presence of organic or inorganic constituents leads to higher oxidant requirements in order to maintain the treatment efficiency [26]. Using biodegradation, the end product (for example via fungal degradation) could be compostable biomass. The majority of amine containing compounds that so far has been successfully biodegraded using fungi with peroxidases are aromatic amines including azo-dyes [27], tannic and humic acid [28], and pharmaceuticals [29]. Aromatic amines can also be adsorbed to sorbents like activated C or modified chitosan [30]. The possibility to biodegrade non-aromatic amines, amides and quaternary ammonium compounds is less investigated.

The perspective of fungi catabolizing hazardous chemical compounds and transforming them to biomass remains understudied and challenging since knowledge is scarce. We hypothesized that fungal species from different ecological groups can survive and grow in the presence of recalcitrant compounds found in wastewaters. Therefor the overall aim of the present study was to test the feasibility of growing fungi for the purpose of metabolizing relevant compounds, utilising part of the fungal species' diversity in Northern European forests and evaluating their growth and survival on N-containing recalcitrant compounds. We wanted to screen a larger number of fungal species with varying taxonomy and ecology, and as a first step we performed a screening study using 48 isolates from 42 species, including both saprotrophic and mycorrhizal fungi. Their ability to survive and grow in high concentration solutions of 2-diethylaminoethanol, $N_3$-trimethyl(2-oxiranyl)methanaminium chloride and 2,6-dichlorobenzamide (BAM) were evaluated. The chemicals are of interest since they are toxic and difficult to handle in wastewater treatments plants. 2-diethylaminoethanol and $N_3$-trimethyl (2-oxiranyl)methanaminium chloride are used as ligands in the large-scale manufacturing of weak and strong anion-exchangers in the protein separation field. BAM is a persistent, water soluble degradation product of the pesticide 2,6-dichlorobenzonitrile (dichlobenil), contaminating ground waters [31]. After the initial screening study a sub-set of species was chosen based on their overall growth on two of the tested compounds (2-diethylaminoethanol and $N_3$-trimethyl(2-oxiranyl)methanaminium chloride) for a simulated process water experiment containing both 2-diethylaminoethanol and $N_3$-trimethyl(2-oxiranyl)methanaminium chloride.

## Materials and methods

### Fungal isolates and experimental systems

Two experiments were set up; firstly a screening experiment (S1 Fig) to test survival and growth of a wide range of fungal species in the presence of 2-diethylaminoethanol, $N_3$-trimethyl(2-oxiranyl)methanaminium chloride, and BAM, and secondly a simulated process water experiment (S2 Fig) including 2-diethylaminoethanol and $N_3$-trimethyl(2-oxiranyl) methanaminium chloride and three fungal species able to grow on the investigated N-containing compounds from the screening experiment. A total of 48 fungal isolates and 42 fungal species (S1 Table) were included in the screening experiment. Within-species variation was tested

for four mycorrhizal species (*Cenococcum geophilum*, *Laccaria laccata*, *Piceirhiza bicolorata*, and *Suillus variegatus*) and two saprotrophic species (*Armillaria mellea* and *Hypholoma fasciculare*). The selection of candidate fungal species for the second experiment was based on their overall growth on two of the tested compounds (2-diethylaminoethanol and N$_3$-trimethyl (2-oxiranyl)methanaminium chloride). Species names, authorities and taxonomical classifications are taken from the Dyntaxa database [32], and information about species ecology from Hallingbäck and Aronsson [33]. The investigated species included both saprotrophic (white rot fungi, brown rot fungi and litter decomposing fungi) and mycorrhizal (ECM and ERM) fungi. Fungal isolates were obtained directly from sporocarps collected from forests around Uppsala in 2005 and from fungal culture collections at the Department of Forest Mycology and Plant Pathology, SLU, Uppsala, Sweden (Petra Fransson and Rimvydas Vasaitis). Obtaining new isolates from sporocarps were done by removing small pieces of fungal tissue from the sterile inside of the sporocarp and placing them on half-strength modified Melin–Norkrans (MMN) medium [34] in 9 cm Petri dishes, until growth was apparent, and fungi were sub-cultured to new plates. All fungal isolates were maintained on MMN medium in darkness at 25˚C and had grown on new plates for one month before starting the experiments. For the screening experiment fungal isolates were grown in Petri dishes in 50 mL basal Norkrans medium [35] with a C:N ratio of 15 and pH adjusted to 4.5. One piece of agar containing mycelia was cut out with a corer (∅ 10 mm) from the actively growing mycelial edge of the fungal culture and placed in the liquid medium (one replicate per species and treatment, with three chemicals and three concentrations, giving a total of 432 plates). Growth controls including basal Norkrans medium only were also prepared (n = 2). In order to increase survival some of the fungi with slow growth rates, mostly ECM species and the ERM fungus *Rhizoscyphus ericae*, were cut out and put on new agar plates for approximately one week so that growth resumed before the agar pieces were transferred to liquid medium. Petri dishes with liquid cultures were incubated in darkness at 25˚C for one week before chemical exposure.

For the simulated process water experiment three fungal species (*Hygrophorus camarophyllus*, *Rhizoscyphus ericae* and *Laccaria laccata* AT2001038) were selected based on growth data in the screening experiment, in combination with how readily the isolates grow in liquid culture (see Results), and were subsequently grown in autoclaved 1000 mL Erlenmeyer flasks containing 200 mL basal Norkrans medium with a C:N ratio of 15 and pH adjusted to 4.5. Ten pieces of agar with actively growing mycelia were initially transferred to each flask (n = 5, giving a total of 15 flasks) with a sterile scalpel. The flasks were sealed with aluminium foil and kept in dark in closed cardboard boxes at room temperature. After one week's growth in basal Norkrans medium 800 mL of the simulated process water was added to each flask (n = 3). Controls (n = 2) containing only 200 mL Basal Norkrans medium were included for each of the three fungal species. The chemicals used in the growth media were supplied from Sigma-Aldrich (Switzerland).

## Chemical exposure and harvest

For the screening experiment standard solutions of 2-diethylamine (Fluka, Switzerland; S1 Fig) and N3-trimethyl(2-oxiranyl)methanaminium chloride (Evonik Industries AG, Germany, trade name; glycidyltrimethylammonium chloride (gly); (S1 Fig) were prepared in autoclaved flasks using autoclaved double distilled water. Both solutions were prepared so that in the screening experiment 2 mL added to a Petri dish with 50 mL liquid medium including mycelia, would give the concentrations 1, 10, and 20 g L$^{-1}$. The choice of these concentrations was based on estimated concentrations of 2-diethylamine and N3-trimethyl(2-oxiranyl)methanaminium chloride (18 and 7.0 g L$^{-1}$, respectively) in a process water at Cytiva, Uppsala, Sweden.

BAM (Acros Organics, Belgium; S1 Fig) was not possible to dissolve at a concentration of 2.7 g L$^{-1}$ as previously reported [36]. A saturated solution was prepared by dissolving 250 mg BAM in one litre warm (80˚ C) double distilled water for 3.5 hours. The undissolved material was removed by vacuum filtration using a 0.45 μm HAWPO4700 cellulose-based Millipore filter. For BAM either 1 μL, 1 mL or 2 mL was added to the Petri dishes. The separate compounds were added to the fungal isolates after a week on liquid medium, during which time the mycelia were adjusted to growing in liquid media and it was assumed that part of the glucose and ammonium sulfate was consumed. After adding the compounds, the fungal isolates were grown for an additional two weeks, giving a total growth period of three weeks. The composition and final concentrations of the simulated process water, chosen to reflect conditions at which the nitrogen containing substances are present in large-scale process water at Cytiva, Uppsala, Sweden, are found in Table 1. To simulate these harsher environments, NaCl and Na$_2$SO$_4$ were added to the 2-diethylaminoethanol and N$_3$-trimethyl(2-oxiranyl)methanaminium chloride, and the C:N ratio in the mixed water was approximately 5:1. The salt concentrations in the process water were determined by inductive coupled plasma mass spectrometry (ICP-MS) analyses of chlorine and sulphur contents at ALS Scandinavia, Luleå, Sweden. The concentrations of the nitrogen containing compounds were estimated by their known consumptions. After addition to the Erlenmeyer flasks, fungi were grown for another three weeks, giving a total growth period of four weeks for the second experiment. The growth controls were harvested already after one week's growth in 200 mL liquid medium. The choice of the different growth periods was motivated by the wish to study the more long-term survival and growth of the selected species under harsh conditions, and their ability to suffice with 2-diethylaminoethanol and N3-trimethyl(2-oxiranyl)methanaminium chloride as their C and N sources. It was thus not prioritized to investigate whether the fungi grew better in basal Norkrans medium or not. At the end of both experiments the contents of the Petri dishes and Erlenmeyer flasks were vacuum-filtered onto weighed filter papers (Munktell 1003, 9 cm). Filter papers were dried in an oven for 24 hrs at 105˚ C and re-weighed for fungal biomass.

## Statistical analysis

Growth of each fungal species in the N-containing compounds' treatments was calculated as a percentage of the mean value of the respective growth control (S2 Table). For the screening experiment differences in the average biomass was tested using a general linear model (GLM) with species and compounds as fixed factors and concentration as covariate, and including the interaction terms species*compounds and species*concentration. The interaction terms compounds*concentration and species*compounds*concentration were removed from the model since they could not be estimated. Pairwise comparisons between species, compounds and species*compounds were done using Tukey method. Further, average biomass in control

**Table 1. Composition of simulated process water and estimated concentrations in a large-scale process water.**

| Process water composition (%) | (w/v)[1] | (w/v)[2] | (w/v)[3] |
|---|---|---|---|
| 2-diethylaminoethanol | 2.4 | 1.9 | 1.8 |
| N$_3$-Trimethyl(2-oxiranyl)methanaminium chloride | 1.0 | 0.8 | 0.7 |
| NaCl | 1.9 | 1.5 | 1.4 |
| Na$_2$SO$_4$ | 0.8 | 0.6 | 0.5 |

[1] Original solution

[2] After 4:1 dilution with Basal Norkrans liquid medium pH was finally adjusted to 4.5

[3] Estimated concentrations in a large-scale process water (Cytiva, Uppsala, Sweden)

treatment between mycorrhizal and saprotrophic fungi, and between types of saprotrophs (white rot, brown rot, and generalists) was tested using one-way analysis of variance (ANOVA), in Minitab 18.1 (Minitab Inc., State College, PA, USA). Ordination analysis was performed using CANOCO version 5.02 (Microcomputer Power, Ithaca, NY, USA). Variation in biomass in controls and all amine treatments (10 response variables) for each fungal isolate (n = 48) was visualized using principal components analysis (PCA), without transforming data. We also used the multi-response permutation procedure (MRPP), a nonparametric procedure in PC-ORD version 5.33 software [37] for testing the hypothesis of no difference between two or more *a priori* assigned groups [38]. This was done to test for the effects of main functional groups (mycorrhizal, saprotroph, saprotroph/parasite, and parasite), functional groups (ECM, ERM mycorrhizal, saprotroph, generalist, white rot, brown rot, litter decomposer, and unknown), phylum (*Ascomycota* and *Basidiomycota*), and order (Agaricales, Atheliales, Boletales, Pezizales, Polyporales, Russulales, and Thelephorales). MRPP provides p-values as well as A-values that measure 'effect sizes,' representing homogeneity within the group compared with that expected randomly. For instance, perfect homogeneity in the group gives A = 1, whereas A values between 0 and 1 indicate that heterogeneity between the groups is greater than that expected by chance. For the simulated process water experiment net growth was calculated by subtracting controls (n = 2) from the total four mean weeks growth including three weeks with added SPW (n = 3).

## Results

### Screening experiment–growth controls

In the control treatment fungi produced on average 24.1 ± 1.6 mg biomass when grown for three weeks in a liquid nutrient media, with somewhat higher biomass (but not significantly so) for mycorrhizal fungi (26.0 ± 2.3 mg) compared to saprotrophic fungi (21.8 ± 2.3 mg). Comparing taxonomic groups within the mycorrhizal fungi the five ascomycetes produced on average 28.5 ± 9.7 mg biomass compared to the 21 basidiomycetes which produced 24.4 ± 3.4 mg. All saprotrophic fungi, with the exception of *Rhizinia undulata*, were basidiomycetes. Comparing the different functional groups and rot types within the saprotrophic fungi, the white rot fungi (11 isolates) produced 25.6 ± 3.2 mg biomass, brown rot (5 isolates) 21.3 ± 11.0 mg, and generalists (2 isolates) 5.5 ± 5.5 mg. The largest biomass was produced by the saprotrophic fungus *Ganoderma applanatum* (62 mg), followed by the mycorrhizal fungi *Pisolithus arhizus* and two isolates of *Piceirhiza bicolorata* (ca. 50 mg) (S2 Table). Some fungal isolates grew poorly in the control treatment (S2 Table); the mycorhizal fungi *Amanita citrina*, *Laccaria laccata* AT2001038, *Rhizoscyphus ericae*, *Thelephora* sp., and *Tricholoma pessundatum* each produced 1.2–6.7 mg biomass (Fig 1), and the saprotrophic fungi *Agaricus arvensis*, *Fistulina hepatica*, and *Fomitopsis pinicola* a comparable 1.2–2.8 mg (Fig 2). For the intra-specific comparisons growth in the control treatment were mostly similar between isolates of the same species (Figs 3 and 4, S2 Table), with the exception for *Laccaria laccata* which varied greatly (2.1 mg and 33.5 mg, respectively).

### General growth responses to N-containing compounds

When fungi were exposed to individual compounds for two weeks most of the 48 isolates were able to survive in liquid media containing amines (S2 Table), and many species were either restricted compared to controls or inhibited. Average biomass production across all treatments was 15.1 ± 0.7 mg, similar between mycorrhizal and saprotrophic fungi (15.9 ± 0.9 mg and 14.2 ± 1.0 mg, respectively), and ranging from no growth (values close to zero; Fig 1H) up to 65 mg (*Schizophyllum commune*, see S2 Table). The biomass values correspond to a growth

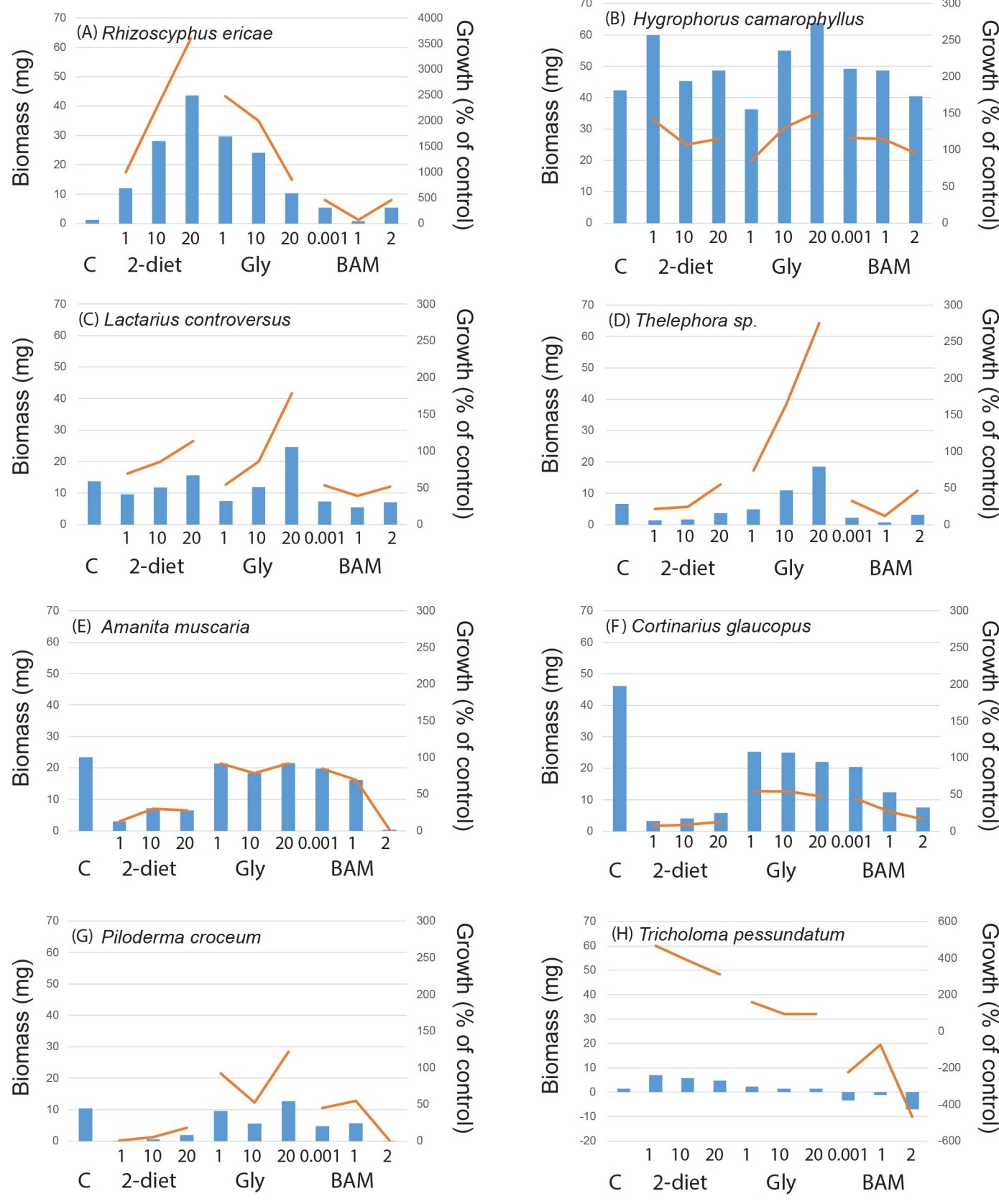

**Fig 1. Mycorrhizal fungal growth responses to recalcitrant amine, amide and ammonium containing media.** Biomass responses to treatments were compared to controls (C, n = 2) for mycorrhizal fungal species in a screening experiment including 2-diethylaminoethanol (2-diet), $N_3$-Trimethyl(2-oxiranyl) methanaminium chloride (gly) and BAM at three different concentrations (n = 1 for each treatment). For 2-diethylaminoethanol and $N_3$-Trimethyl(2-oxiranyl) methanaminium chloride concentrations were 1, 10, and 20 g $L^{-1}$, respectively. For BAM 1μL, 1 mL, and 2 mL of a saturated solution was added. Bars show biomass (mg), lines show growth as a percentage of the growth controls. Mycorrhizal species showed positive growth responses to all amine treatments exemplified by (A) *Rhizoscyphus ericae* and (B) *Hygrophorus camarophyllus*, positive growth responses to some amine treatments exemplified by (C) *Lactarius controversus* and (D) *Thelephora* sp., negative responses to all treatments (E) *Amanita muscaria*, and (F) *Cortinarius glaucopus*. The sample (G) represents a commonly occurring species in boreal forests (*Piloderma crocuem*), and (H) *Tricholoma pessundatum* exemplifies a species which produced little biomass in the control and amine treatments.

increase relative the controls up to a 36-fold increase (S2 Table, Figs 1–4). For some treatments where the final biomass production was close to zero at harvest, the biomass from the first week of growth on basal Norkrans medium decreased when exposed to the selected compounds. The GLM showed that there were significant overall effects of species (F = 13.27, P<0.0001) and compounds (F = 79.12, P<0.0001) on biomass, as well as a significant interaction between species and compounds (F = 1.50, P = 0.003), but no effect of compound concentration. The model explained 72.25% of the biomass variation. Overall fungi grown in 2-diethylaminoethanol (lowest average biomass) and BAM produced significantly less biomass than in both control and $N_3$-Trimethyl(2-oxiranyl)methanaminium chloride (similar average biomass for the latter two). Pairwise comparisons for the interaction term species and compounds are shown in S3 Table. In general, there were more negative growth responses to all three compounds than positive (S2 Table). Biomass production was positively affected by all three compounds for the mycorrhizal *Rhizoscyphus ericae* (up to 36-fold growth increase; Fig 1A) and *Hygrophorus camarophyllus* (up to 1.5-fold; Fig 1B), as well as for the saprotrophic *Fomitopsis pinicola* (up to 15-fold increase; Fig 2A), *Mycena epipterygia* (up to 4.5-fold Fig 2B) and *Rhizina undulata* (up to 1.5-fold Fig 2H). Negative effects by all three N-containing compounds at all three concentrations compared to controls were found for eight mycorrhizal fungi (*Amanita muscaria*, *Cortinarius glaucopus*, *Hebeloma* sp. 1, *Laccaria bicolor*, *Paxillus involutus*, *Piloderma byssinum*, *Suillus bovinus*, and *Suillus luteus*; S2 Table, Fig 1E and 1F), and for three saprotrophic fungi (*Ganoderma applanatum*, *Lenzites betulina*, and *Hypholma* sp.; S2 Table, Figs 2E, 2F and 4E). The phylogenetic distribution of the 48 fungi plotted against the growth responses to the three highest compound concentrations showed that the ability for fungal isolates to increase growth in the presence of the substances varies across the range of systematic entities and species (S3 Fig). Hence, no single evolutionary group seem to have a clear advantage in biodegrading these compounds, but rather that it appears to be important to apply an evolutionary broad screen when selecting suitable taxa. The multivariate analysis revealed no patterns in growth response for the different functional groups (Fig 5), neither for the main groups nor for the detailed rot types etc. within the saprotrophic fungi. For systematic levels, the MRPP analyses showed significant differences between the two phyla *Basidiomycota* and *Ascomycota* (MRPP analysis; p = 0.045, A = 0.025), and between fungal orders (MRPP analysis; p = 0.0067, A = 0.093). The average biomass responses for each species to both controls and all treatments are shown in S4 Fig.

## Growth responses to specific N-containing compounds

There were some general response patterns for growth on the individual compounds (see S2 Table). For 2-diethylaminoethanol 19 out of 26 mycorrhizal isolates and 16 out of 22 saprotrophic isolates showed a negative growth response for all concentrations. For BAM 15 mycorrhizal fungi and 15 saprotrophic fungi were negatively affected by all concentrations, followed by $N_3$-trimethyl(2-oxiranyl)methanaminium chloride where eight mycorrhizal and three saprotrophic fungi were negatively affected. Among the mycorrhizal fungi, positive growth

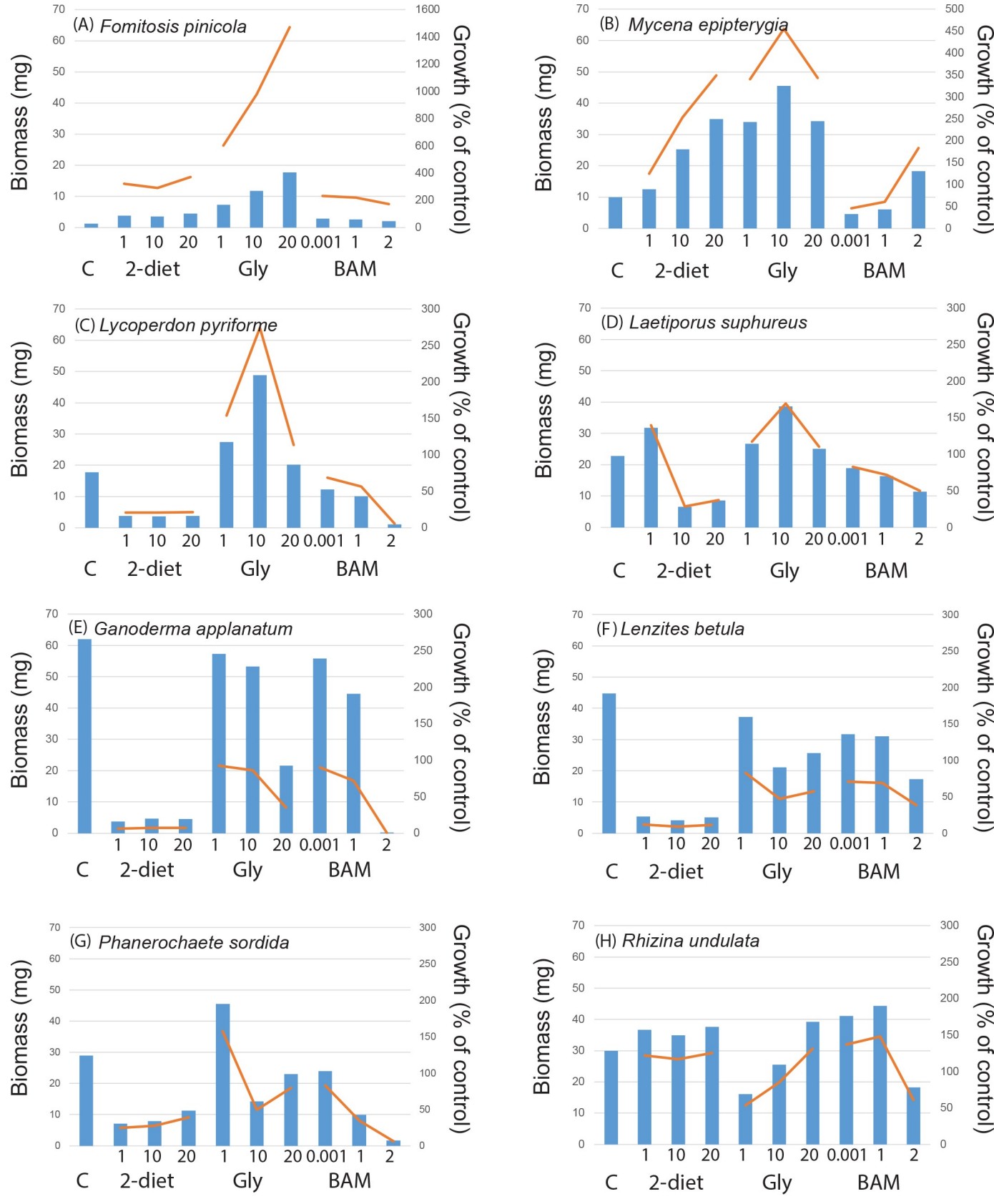

**Fig 2. Saprotrophic fungal growth responses to recalcitrant amine, amide and ammonium containing media.** Biomass responses to treatments as compared to controls (C, n = 2) for saprotrophic fungal species in a screening experiment including 2-diethylaminoethanol (2-diet), $N_3$-Trimethyl(2-oxiranyl)methanaminium chloride (gly) and BAM at three different concentrations (n = 1 for each treatment). For 2-diethylaminoethanol and $N_3$-Trimethyl(2-oxiranyl)methanaminium chloride concentrations were 1, 10, and 20 g $L^{-1}$, respectively. For BAM 1μL, 1 mL, and 2 mL of a saturated solution was added. Bars show biomass (mg), lines show growth as a percentage of the growth controls. Saprotrophic species showed positive growth responses to all amine treatments exemplified by (A) *Fomitopsis pinicola* (brown rot) and (B) *Mycena epipterygia* (litter decomposer). Two brown rot fungi with positive growth responses to some amine treatments were exemplified by (C) *Lycoperdon pyriforme*, and (D) *Laetiporus suphureus*. Negative growth responses to all treatments were exemplified by (E) *Ganoderma applanatum* (brown rot) and (F) *Lenzites betula* (white rot), as well as two parasites (G) *Phanerochaete sordida*, and (H) *Rhizina undulata*.

responses (for one or more concentrations) were most common when grown on $N_3$-trimethyl (2-oxiranyl)methanaminium chloride; 13 mycorrhizal species (incl. *Amanita muscaria*, *Cenococcum geophilum*, *Hygrophorous amarophyllus*, both isolates of *Laccaria laccata*, *Lactarius controversus*, *Leccinum scabrum*, *Piloderma croceum*, *Pisolithus arhizus*, *Rhizopogon roseolus*, *Rhizoscyphus ericae*, *Suillus variegatus* (1st Sept 04), *Thelephora* sp., and *Tricholoma pessundatum*). Similarly, for saprotrophic fungi positive growth responses were most common when grown on $N_3$-trimethyl(2-oxiranyl)methanaminium chloride; 19 fungi with the exception of *Ganoderma applanatum* and *Hypholoma* sp. When comparing the intra-specific variation in biomass and growth responses to the N-containing compounds, patterns were very similar for all species except *Laccaria laccata* (Figs 3 and 4).

## Simulated process water experiment

Based on growth data in the experiment, in combination with how easy the isolates were to grow in liquid culture, the mycorrhizal species *Rhizoscyphus ericae*, *Hygrophorus camarophyllus*, and *Laccaria laccata* AT2001038 were chosen for the simulated process water experiment. All three isolates survived and grew on the simulated process water (Fig 6). Net biomass and growth responses corresponded to 179 mg (380%), 292 mg (136%) and 336 mg (292%) for *Laccaria laccata*, *Hygrophorus camarophyllus*, and *Rhizoscyphus ericae*, respectively.

## Discussion

In a first experiment, the growth and survival of 48 fungal isolates with varying taxonomy and ecology on three different N-containing compounds (2-diethylaminoethanol, $N_3$-trimethyl (2-oxiranyl)methanaminium chloride, and BAM) at three concentrations were evaluated. The isolates belonged to the two main functional groups saprotrophic and mycorrhizal fungi, which are known for their complex enzyme systems used for depolymerizing organic matter [5, 8], ability to compete for nutrients in soil and woody substrates, as well as being relatively easy to grow in pure culture. We confirmed our hypothesis, that both saprotrophic and symbiotic fungal species can survive and grow in the presence of recalcitrant compounds found in wastewaters. Although many isolates were partly restricted or inhibited in growth in the presence of the selected substances, most survived. A subset of three mycorrhizal isolates, which were further tested in a simulated process water experiment, produced large biomass despite exposure to harsh conditions similar to those at which the compounds are present in large-scale manufacturing plants. In this study, we are predicting fungal ability to degrade the recalcitrant N compound by their growth and biomass. However, this first screening study needs to be followed by more in-depth studies confirming decreased concentrations of these substances.

## Do fungal functional groups differ in their responses to individual N-containing compounds?

Comparing the main functional groups, both mycorrhizal and saprotrophic fungal isolates were able to produce similar amounts of biomass when grown in control treatments, and

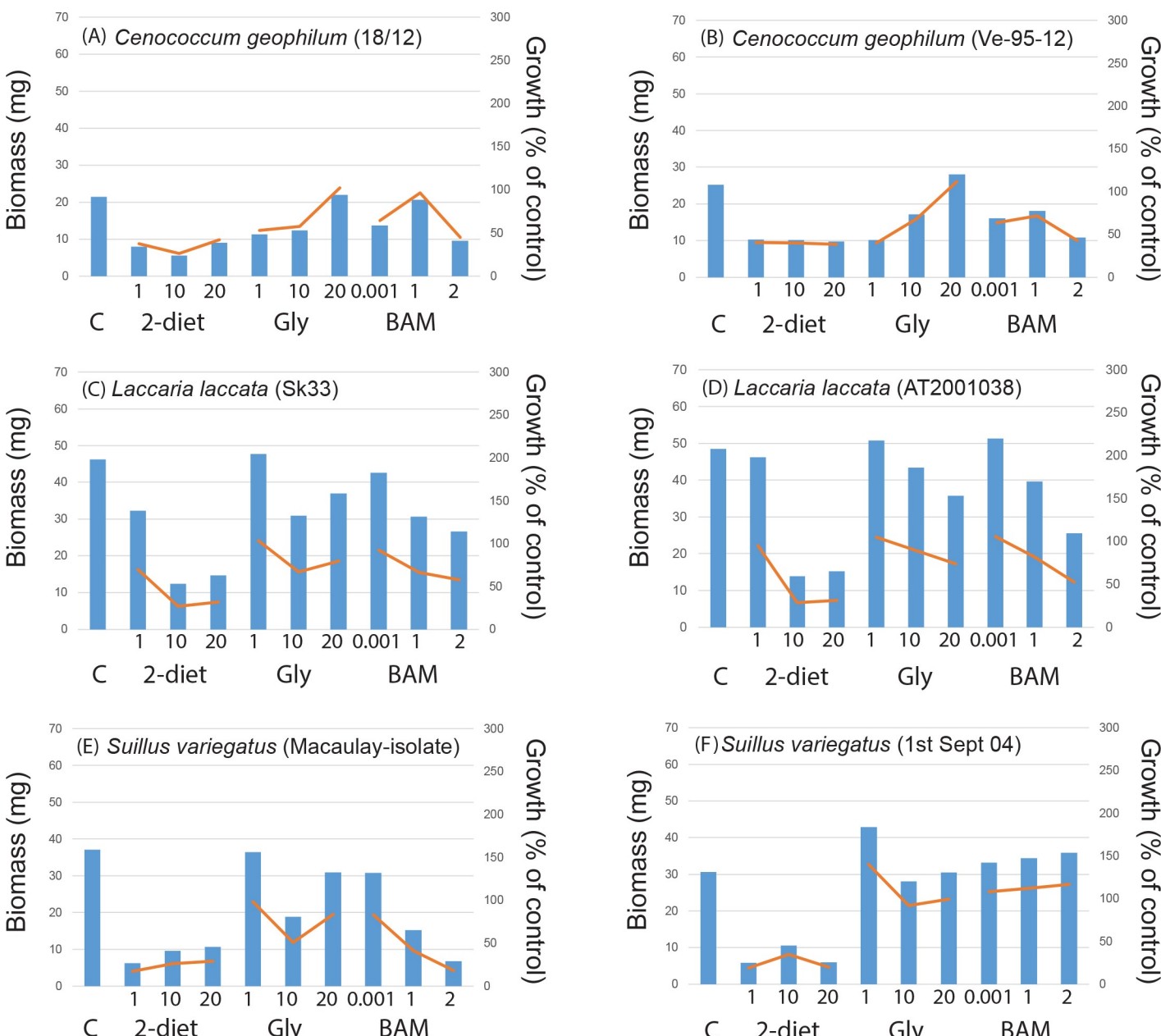

**Fig 3. Intra-specific mycorrhizal fungal growth responses to recalcitrant amine, amide and ammonium containing media similar among isolates.** The intra-specific variation (n = 2 isolates per species) in biomass responses to treatments compared to controls (C, n = 2) for mycorrhizal fungal species in a screening experiment including 2-diethylaminoethanol (2-diet), $N_3$-Trimethyl(2-oxiranyl)methanaminium chloride (gly) and BAM at three different concentrations (n = 1 for each treatment). For 2-diethylaminoethanol and $N_3$-Trimethyl(2-oxiranyl)methanaminium chloride concentrations were 1, 10, and 20 g $L^{-1}$, respectively. For BAM 1μL, 1 mL, and 2 mL of a saturated solution was added. Bars show biomass (mg), lines show growth as a percentage of the growth controls: (A) and (B) *Cenococcum geophilum*, (C) and (D) *Laccaria laccata*, (E) and (F) *Piceirhiza bicolorata*, and (G) and (H) *Suillus variegatus*, respectively.

among the saprotrophic fungi wood decomposing species with and without peroxidases (white and brown rot fungi, respectively) tended to grow better than the few species that are generalists. There was large inter-specific variation in growth among the tested isolates, which is in line with earlier studies conducted in pure culture [39, 40]. Low biomass production in some isolates may reflect slow growth rates for some species when grown in pure culture or

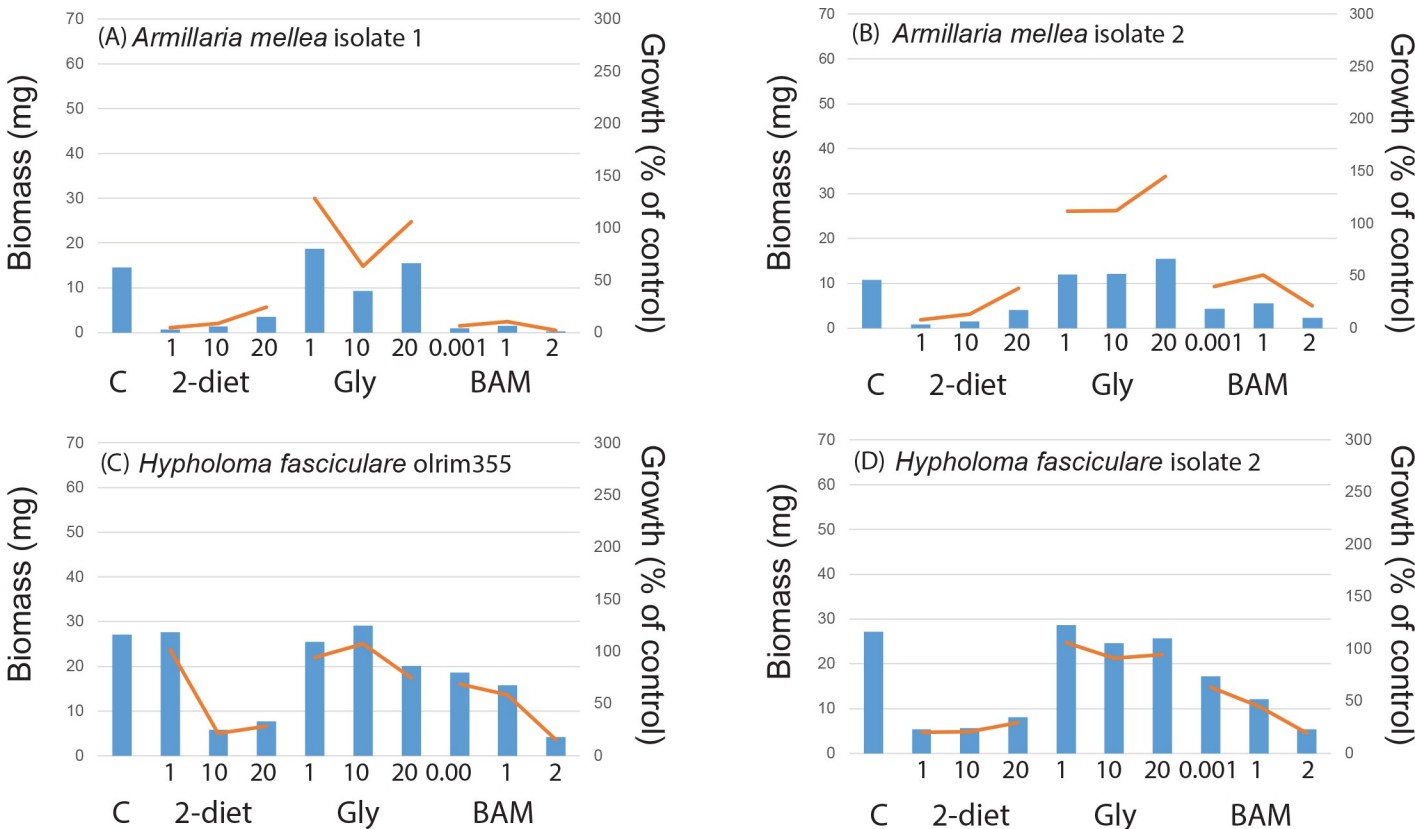

**Fig 4. Intra-specific saprotrophic fungal growth responses to recalcitrant amine, amide and ammonium containing media similar among isolates.** Intra-specific variation in biomass responses to treatments compared to controls (C, n = 2) for saprotrophic fungal species in a screening experiment including 2-diethylaminoethanol (2-diet), $N_3$-Trimethyl(2-oxiranyl)methanaminium chloride (gly) and BAM at three different concentrations (n = 1 for each treatment). For 2-diethylaminoethanol and $N_3$-Trimethyl(2-oxiranyl)methanaminium chloride concentrations were 1, 10, and 20 g $L^{-1}$, respectively. For BAM 1μL, 1 mL, and 2 mL of a saturated solution was added. Bars show biomass (mg), lines show growth as a percentage of the growth controls: (A) and (B) *Armillaria mellea*, (C) and (D) *Hypholoma fasciculare*, respectively.

indicate use of an unsuitable substrate for other isolates. Although many species were partly restricted or inhibited in growth, most survived when the investigated compounds were added. This indicates an ability to utilize the compounds as substrates, and a large biomass was assumed to indicate fungal use via either enzymatic biodegradation, biosorption or bioaccumulation. In a previous study including 44 fungal isolates from vineyard soil and grapevine the ability to degrade biogenic amines was noteworthy for many fungi, and independent of the amine incorporated into the culture medium [41]. In the present study, mycorrhizal fungi showed generally more negative responses to all three N-containing compounds compared to the saprotrophic fungi, which probably reflects a higher ability of for example wood decomposers (white rot fungi) to tolerate toxic chemicals and environments within e.g. wood [42]. The non-specific degradation mechanisms using extracellular enzymes, allow lignolytic fungi to degrade a wide range of recalcitrant pollutants [1, 43–45]. Despite this general pattern when comparing responses to all three individual substances, some mycorrhizal isolates also coped well. For example, suilloid species (*Suillus* spp. and *Rhizopogon roseolus*) are well known to produce large amounts of biomass (e.g. [39]) and do so when exposed to amines, and these results were confirmed in the present screening study. Among the mycorrhizal species included in the study, we only had one isolate of an ascomycete forming ERM mycorrhiza (*Rhizoscyphus ericae*), which was chosen for the simulated process water experiment due to a

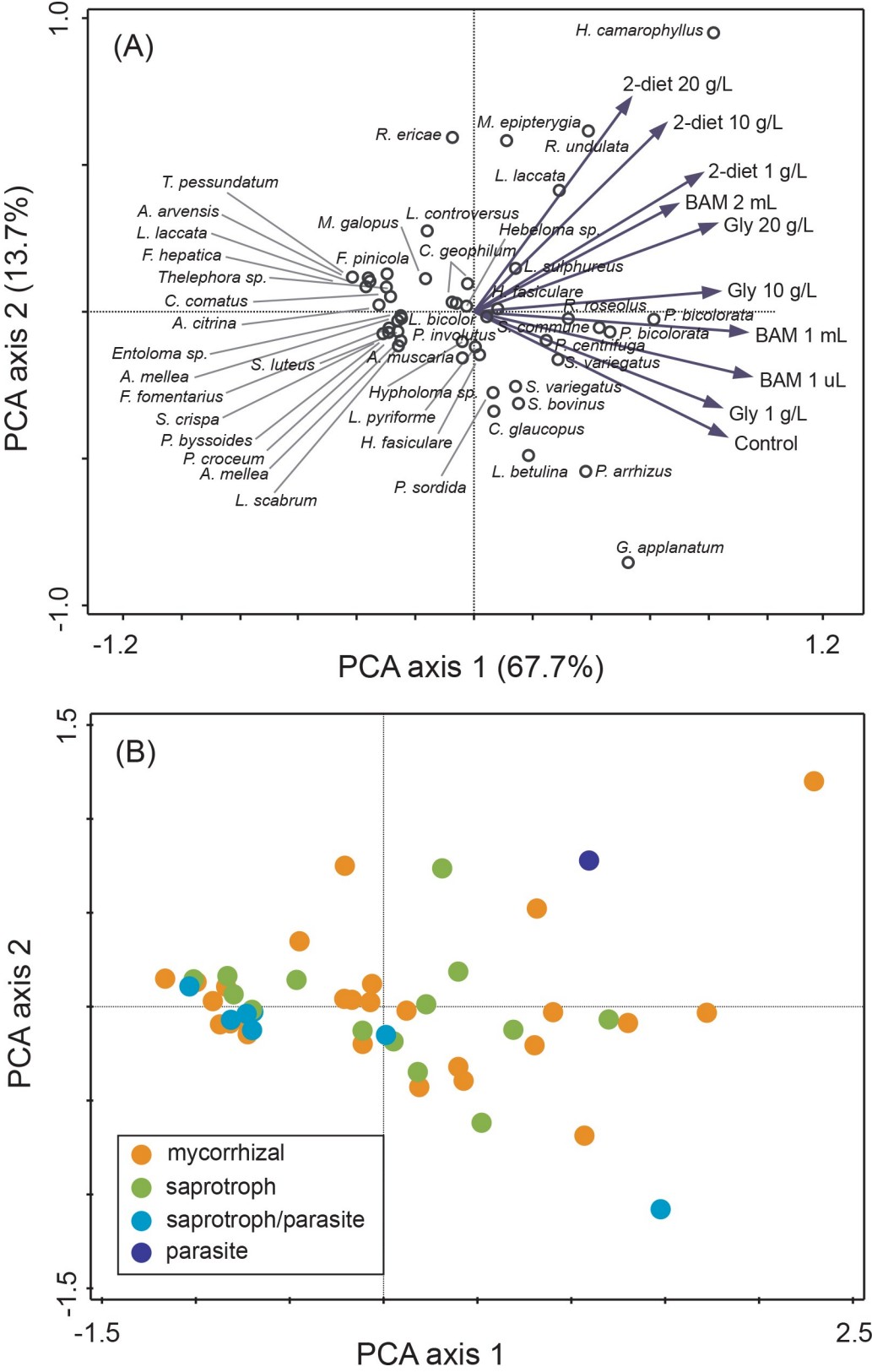

**Fig 5. Overall fungal biomass responses to the three tested N-containing recalcitrant compounds in the screening experiment.** Principal component analysis (PCA) showing the variation in biomass responses for 48 fungal isolates when

grown for three weeks in control treatment with nutrient solution and N-containing compounds' treatments (2-diethylaminoethanol [2-diet], $N_3$-Trimethyl(2-oxiranyl)methanaminium chloride [gly] and BAM) at three different concentrations, respectively. For 2-diethylaminoethanol and $N_3$-Trimethyl(2-oxiranyl)methanaminium chloride concentrations were 1, 10, and 20 g $L^{-1}$. For BAM 1μL, 1 mL, and 2 mL of a saturated solution was added. Species differences are visualized by (A) a sample plot with the vector length indicating the relative importance of the amine treatments, and (B) a sample plot with species coded according to functional groups. The first three axes together explained 87.5% of the total variation (84181.4).

strongly positive biomass response to the N-containing compounds. This species belongs to an aggregate of species [46] also including *Piceirhiza bicolorata* with yet unclear systematic affinities, which can form ECM associations. The species aggregate is of special interest in the context of withstanding or metabolising N-containing compounds of the type included in our study, since they possess a wide range of biochemical and physiological attributes enabling the fungus to cope with the harsh and stressed habitats of ericoid plants [47]. It was clear from the screening experiment that all three isolates from this species aggregate produce very large amounts of biomass in the current set-up. Experimental studies have confirmed their saprotrophic capabilities [48] with a wide range of extracellular enzymes, and they are known to utilize ammonium, nitrate, organic substances like amino acids [49] and their amides [50], and proteins [51]. Further, *Rhizoscyphus ericae* is able to mobilize organic N also from even more recalcitrant sources such as lignin [52] and chitin [53, 54]. *Laccaria laccata*, one of the ECM species included in the simulated process water experiment, is known to be easy to grow in liquid culture [55] and has shown potential as biological control agent against disease causing fungi [56, 57]. Although ECM fungi such as *Laccaria laccata* degrade pollutants and expedite removal of persistent organic pollutants [58, 59], it is unknown whether the species can metabolise amines, amides or quaternary ammonium compounds. However, the closely related species *Laccaria bicolor* was previously shown to be unable to grow on media containing amines as sole N sources [60] and is suggested in nature to use the ammonium produced either by microbial or chemical amine decomposition since it has been shown to have little or no ability to grow on organic N sources [61]. The main conclusion of the growth experiment including either 2-diethylaminoethanol, $N_3$-trimethyl(2-oxiranyl)methanaminium chloride,

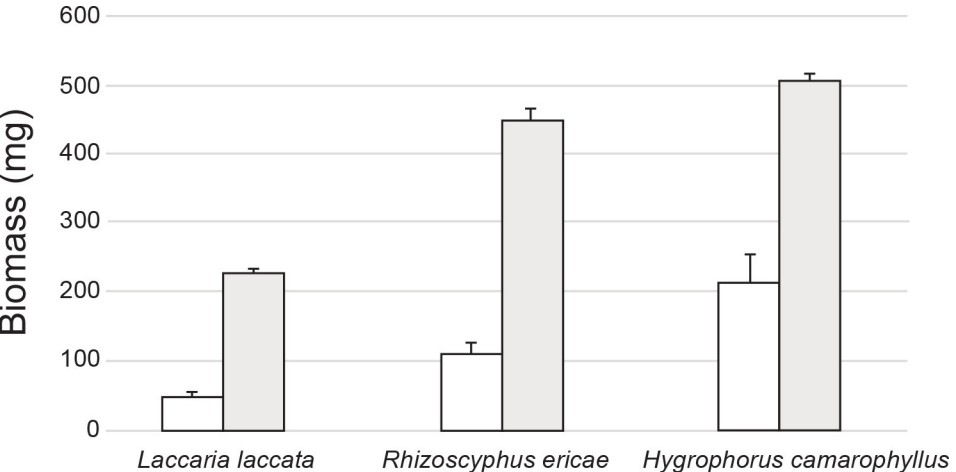

**Fig 6. Biomass production for three mycorrhizal fungal species grown for three weeks in simulated process water.** Average biomass production ± stdev for controls grown for one week in Basal Norkrans medium (white bars; n = 2), and the simulated process water treatment grown for one initial week in Basal Norkrans medium followed by three weeks in amine solution (grey bars; n = 3).

or BAM was that many fungal isolates survived and grew in the presence of these N-containing compounds.

## Simulated process water experiment

In the simulated process water, all three tested species grew well, and the addition of salts did not seem to significantly prevent their growth. The fungi most likely continued to use the Basal Norkrans medium as nutrients in the three weeks period including the simulated process water, however, the excess of test solution in combination with the large biomass indicated that the fungi used the N-containing compounds in the process waters as substrates for growth. *Rhizoscyphus ericae* produced the largest biomass (ca. 340 mg) and *Laccaria laccata* showed the highest growth increase. The observed growth of the fungi on the N-containing substrates was most likely explained by either extracellular or endo-enzymatic degradation mechanisms. In the first case, the products from the biodegraded compounds must penetrate the fungal cell bi-layers, and in the endo-enzymatic mechanism, the native substances are transported through the membranes for further degradation within the cells. Several kinds of filamentous fungi are known to produce amine oxidase activity when using amines as a sole N source for growth [62–64]. Two kinds of amine oxidases were the first to be purified and characterized from fungi [65, 66], later followed by studies revealing other types of amine oxidases (e.g. [67]). The enzymes catalyze the oxidative deamination of terminal amino groups, allowing the fungi to degrade an amine as a source of ammonium for growth. This would explain the ability of many fungal isolates to increase biomass in the presence of amines, since N often is the most growth-limiting nutrient. In the present study, the C:N ratio of the simulated process water was low (5:1) and most of the N was not present in a directly available form, thus the fungi must have the ability to metabolize the selected substrates to promote uptake and biomass production. This, however, needs to be confirmed by for example analyzing residual N-containing compounds in the liquid media or investigating the potential presence of amine oxidases and other relevant enzymes than can catalyze the investigated compounds. Amine oxidase activity was first observed in strains of *Aspergillus niger*, *Aspergillus fumigatus*, *Penicillium chrysogenum* and *Penicillium notatum* [65], which are well-known representatives of the order Eurotiales in *Ascomycota*. In a more recent study evaluating the ability of vineyard soil and grapevine fungi to degrade biogenic amines *Penicillium* spp., *Alternaria* sp., *Phoma* sp., *Ulocladium chartarum* and *Epicoccum nigrum* showed high capacity to *in vitro* amine degradation in a microfermentation system [41]. These are also species within *Ascomycota*, where all (with the exception of the genus *Penicillium*) belong to the order Pleosporales. In the present study we did not include any species from these orders, since we focused mainly on fruitbody forming saprotrophic and mycorrhizal fungi belonging to *Basidiomycota*, with a few exceptions found in *Ascomycota*. The species included here represent other ecological groups of fungi compared to the examples from Pleosporales and Eurotiales. Amine oxidase activity was previously detected in one basidiomycotous species, which is also included in the present study, *Armillaria* (saprotroph/parasite), in a large screening study investigating 85 fungal isolates [66], along with a number of species belonging to *Ascomycota*. Beside from these studies, little is known about the distribution of the enzyme systems in fungal strains from different ecosystems, and as far as we are aware, it is unknown whether amino oxidases are present in most saprotrophic or mycorrhizal fungi. In future studies, it would be of interest to design the experiments so that concentration changes in N-containing compounds can be measured, requiring lower substrate amounts.

In summary, the feasibility of growing fungi for metabolizing recalcitrant N containing compounds, including an amine, an amide, and a quaternary compound, from a simulated

wastewater was tested, utilising part of the large fungal species diversity in Northern European forests. The species included in the present study differed from earlier studies of filamentous fungi in the context of e.g. amine oxidation of these substances, since they belong to the functional groups wood and litter decomposers, and mycorrhizal fungi. Although many isolates were partly restricted or inhibited in growth, most survived in the presence of 2-diethylethanolamine, $N_3$ trimethyl(2-oxiranyl)methanaminium chloride, and BAM. The observed growth on these compounds is to our knowledge not previously reported and confirmed the hypothesis that both saprotrophic and symbiotic fungal species can survive and grow in their presence. The most promising fungi of those tested, when growth data were considered, was the ECM fungus *Laccaria laccata* and the ERM mycorrhizal fungus *Rhizoscyphus ericae*. In addition to the saprotrophic fungi, especially fungi with peroxidases, which are used in whole cell fungal treatments within industry, mycorrhizal fungi showed potential as alternatives for treatments of wastewater containing the investigated N containing substances. However, this first screening study needs to be followed by more in-depth studies confirming decreased concentrations of these substances.

## Supporting information

**S1 Fig. Experimental scheme for the screening experiment.** A total of 48 fungal isolates where grown in Petri dishes for a total of three weeks in liquid growth media containing individual recalcitrant compounds. Concentrations for the N-containing compounds were 1 g L$^{-1}$, 10 g L$^{-}$1, and 20 g L$^{-1}$ for 2-diethylaminoethanol and $N_3$-Trimethyl(2-oxiranyl)methanaminium chloride, and for 2,6-dichlorobenzamide (BAM) 1 μL, 1 mL, and 2 mL were added from a saturated solution. No substance was added to the growth controls. The recalcitrant N-containing compounds are depicted to the right.
(TIF)

**S2 Fig. Experimental scheme for the simulated process water experiment.** Three mycorrhizal fungal species (*Hygrophorus camarophyllus*, *Rhizoscyphus ericae*, and *Laccaria laccata*) were grown for a total of four weeks in Erlenmeyer flasks containing a recalcitrant amine/amide mixture. For composition of mixture see Table 1.
(TIF)

**S3 Fig. The fylogenetic distribution of 48 fungal isolates included in the screening experiment, plotted against the growth responses.** The growth responses represent the isolates (% of controls) containing the three highest N-containing compounds' concentrations. Concentrations corresponded to 20 g L$^{-1}$ for 2-diethylaminoethanol (blue bars) and $N_3$-Trimethyl (2-oxiranyl)methanaminium chloride (orange bars), and addition of 2 mL saturated 2,6-dichlorobenzamide (BAM) solution (red bars). Values over 100% means that fungi grew better with the amines present. The red branches are ECM fungi, the blue saprotrophs and the green ERM mycorrhiza. The diagram was cut at 500%, missing values and negative values were set to 0%.
(TIF)

**S4 Fig. Principal component analysis (PCA) showing the overall variation in fungal biomass responses to three recalcitrant N-containing compounds.** 48 fungal isolates when grown for three weeks in control treatment with nutrient solution and amine treatments (2-diethylaminoethanol [2-diet], $N_3$-Trimethyl(2-oxiranyl)methanaminium chloride [gly] and 2,6-dichlorobenzamide [BAM]) at three different concentrations, respectively. For 2-diethylaminoethanol and $N_3$-Trimethyl(2-oxiranyl)methanaminium chloride concentrations were 1, 10, and 20 g L$^{-}$1. For BAM 1μL, 1 mL, and 2 mL of a saturated solution was added. Species

differences are visualized by a sample plot with circle size depicting the average biomass response across all treatments. The first three axes together explained 87.5% of the total variation (84181.4).
(TIF)

**S1 Table. Mycorrhizal and saprotrophic fungal isolates.** The fungal isolates were used to screen for survival and growth in liquid media containing recalcitrant amine, amide and ammonium compounds.
(XLSX)

**S2 Table. Biomass production by 48 fungal isolates grown for three weeks in liquid culture in the presence of three individual recalcitrant N-containing compounds.**
(XLSX)

**S3 Table. Pairwise comparisons between species and compounds.** For the screening experiment differences in the average biomass for 48 fungal isolates grown in control treatment with nutrient solution and three amine treatments was tested using a general linear model and Tukey method, here reporting the interaction term species and compounds.
(XLSX)

## Acknowledgments

We are grateful to Dr. Rimvydas Vasaitis at Department of Forest Mycology and Plant Pathology for supplying fungal isolates, and Dr. Robert Burman, Medical Products Agency, Uppsala, Sweden for his contribution in the simulated process water experiment.

## Author Contributions

**Conceptualization:** Åke Stenholm, Anders Backlund, Sara Holmström, Petra Fransson.

**Data curation:** Åke Stenholm, Anders Backlund, Sara Holmström, Petra Fransson.

**Formal analysis:** Åke Stenholm, Anders Backlund, Sara Holmström, Maria Backlund, Petra Fransson.

**Investigation:** Sara Holmström, Maria Backlund.

**Methodology:** Åke Stenholm, Anders Backlund, Sara Holmström, Maria Backlund, Petra Fransson.

**Project administration:** Anders Backlund, Petra Fransson.

**Resources:** Anders Backlund, Petra Fransson.

**Supervision:** Anders Backlund, Mikael Hedeland, Petra Fransson.

**Validation:** Åke Stenholm, Anders Backlund, Sara Holmström, Maria Backlund, Mikael Hedeland.

**Visualization:** Åke Stenholm, Anders Backlund, Petra Fransson.

**Writing – original draft:** Anders Backlund, Petra Fransson.

**Writing – review & editing:** Åke Stenholm, Anders Backlund, Sara Holmström, Maria Backlund, Mikael Hedeland, Petra Fransson.

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
