## [Decision Letter · Decision Letter 0]

23 Mar 2021

PONE-D-20-39694

Survival and growth of saprotrophic and mycorrhizal fungi in recalcitrant amine, amide and ammonium containing media

PLOS ONE

Dear Dr. Fransson,

Thank you for submitting your manuscript to PLOS ONE. After careful consideration, we feel that it has merit but does not fully meet PLOS ONE’s publication criteria as it currently stands. Therefore, we invite you to submit a revised version of the manuscript that addresses the points raised during the review process.

Reviewers comments on your MS are now received. Your MS needs major corrections before we may consider the MS for publication in PLSO One. Kindly do the needful corrections and submit your MS with a point-wise response. 

We look forward to receiving your revised manuscript.

Kind regards,

Vijai Gupta, PhD in Microbiology

Academic Editor

PLOS ONE

Journal Requirements:

We note that one or more of the authors are employed by a commercial company: "Cytiva".

Additional Editor Comments (if provided):

Reviewers comments on your MS are now received. Your MS needs major corrections before we may consider the MS for publication in PLSO One. Kindly do the needful corrections and submit your MS with a point-wise response.

Reviewers' comments:

Reviewer's Responses to Questions

**Comments to the Author**

1. Is the manuscript technically sound, and do the data support the conclusions?

Reviewer #1: Yes

Reviewer #2: Yes

2. Has the statistical analysis been performed appropriately and rigorously? 

Reviewer #1: Yes

Reviewer #2: No

3. Have the authors made all data underlying the findings in their manuscript fully available?

Reviewer #1: Yes

Reviewer #2: Yes

4. Is the manuscript presented in an intelligible fashion and written in standard English?

Reviewer #1: Yes

Reviewer #2: Yes

5. Review Comments to the Author

Reviewer #1: The present study entitled “Survival and growth of saprotrophic and mycorrhizal fungi in recalcitrant amine, amide and ammonium containing media” assessed fungal diversity from Northern European forests for metabolizing recalcitrant N containing compounds, including an amine, an amide and a quaternary compound, in wastewater.

In general, the manuscript is nicely written, easy to read and mostly free of formal flaws.

However, there are some points that needs to be addressed before final publication. I have some comments which may be useful for improving the original version of this paper:

1. What a bar represents above column in figure 6? Is this standard error or standard deviation. Kindly provide the detail.

2. Hypothesis and objectives of the present study are not mentioned anywhere in the manuscript. Explain them while concluding the introduction section.

3. Why author has selected only three species i.e. Rhizoscyphus ericae, Hygrophorus camarophyllus, and Laccaria laccata AT2001038 in simulated process water experiment while some other species are also performing well in screening experiment.

4. The composition and final concentrations of the simulated process water, chosen to reflect the conditions at which the amines are present in large-scale manufacturing plants. It would be great if author may provide some reference data in separate column in table 1 to represent the concentration of these N compounds in different manufacturing plants.

5. The results of this study may be useful for the treatments of industrial wastes. To consider these species as successful biological tool for the degradation of various hazardous nitrogenous compound only on the basis of biomass production is not sufficient and appropriate. However, I know this is just a preliminary study and future advancement in analysing the initial and final concentration of N containing compounds in liquid media and various enzymes responsible for the degradation of these compounds such as amine oxidases would provide direct evidence of catalysing ability of these microbes for investigating compounds.

6. At some places in manuscript, concentration of various recalcitrant N compounds is provided as g L-1 or g/L, which needs uniformity.

7. In figure 1 to 4, lowest concentration of BAM is provided in µL while in supplementary excel file (S2 table1) unit of BAM is given in ml. Kindly check and correct it accordingly.

8. In reference section, some of the journal names are abbreviated while others are explained fully. Kindly check and revise thoroughly.

9. Kindly explain all the abbreviation fully when they appear first time such as ERM.

Reviewer #2: This preliminary in vitro screening study examines the potential of several saprophytic and mycorrhizal fungi originating from the Northern European forests to tolerate and grow on media containing amide, amine, and ammonium compounds. Moreover, three mycorrhizal fungal isolates with presumably superior performance in the screening experiment were tested for growth in stimulated water experiment spiked with amine and quaternary ammonium compounds. The results of the study indicate that Rhizoscyphus ericae performed better in the screening experiment whereas Laccaria laccata accumulated increased biomass under-stimulated realistic conditions. Form these results it was concluded that some of the fungal isolates examined could be potential candidates for remediating wastewater contaminated with these recalcitrant chemicals. Although the results are of interest, there are some concerns as mentioned below that need attention.

1. The concentrations of the nitrogenous compounds (1, 10, 20 g/L) used in the screening study is the major concern in the study. It is not clear how concentrations of these compounds are deduced. Was there any trial run performed to fix the minimum and maximum limit for these compounds?

2. The second concern of the study is that the concentration of certain compounds in screening was not determined. The quantity of the BAM present in the experimental solutions is not known as some of the undissolved portions were removed from the solution?

3. The rationale for selecting mycorrhizal fungi for stimulated water processing is a bit confusing (Lines 187–190) as mycorrhizal fungi in general exhibited more negative responses than the saprophytic fungi (Lines 422–424). Further, there were saprophytic fungi that had a more positive response to biomass accumulation in the different compound concentrations than Laccaria lacata. It is better to consider the average response of a fungal species to all the exposed chemicals at different concentrations for selection rather than the response to the individual concentration of a chemical.

4. Please indicate how the fungal isolates obtained directly from field sampling for the study were authenticated. Moreover, mention the accession numbers of the isolates used from the fungal culture collection. At present some codes (these are not accession numbers of the validated isolates?) are mentioned for some taxa in the supplementary table, and it is missing for many.

5. Lines 221–222: Justify the reason for harvesting controls after one week of growth. As the fungal growth under any conditions varies with time, compare samples from similar time points. How could you compare the growth of one-week-old fungal culture (control) with four-week-old fungal cultures (treatments)?

6. The statistical approach used in the study needs rethinking. For instance, in the screening experiment, there were 48 fungal isolates, three experimental compounds, and different concentrations (including control). Though one-way ANOVA could bring out the variation among the fungal isolates for a tested compound, the reactivity of a fungal isolate to the different compounds and their concentrations tested is obscure. In such a condition, a three-way ANOVA instead of one-way ANOVA would be more meaningful as it could bring out the significance in the variation not only between species but also among compounds and their concentrations. Further, perform Post Hoc analysis for variances that are significant and present the results of post hoc analysis in the figures and tables. As the treatments are compared with a single control, Dunnett’s test would be more appropriate.

7. The introduction is a bit long. Concise the introduction by removing facts that are too basic and well-known. Further, it is not clear how the detailed information on the different types of fungi (saprophytic and ectomycorrhizal) to do with the present study. Sum up all this information into one or few s sentences.

8. Some of the results presented do not agree with the data presented in the tables. For example, Line 263 states that all the saprophytic fungi were basidiomycetes. However, Rhizinia undulata examined in the present study (supplementary table 1) is an ascomycetous fungus.

9. Figures should be self-explanatory. Instead, details in the figures are missing in the legends presented in the results (insertions). For example, in figures (1-4) treatment names in the x-axis are abbreviated, and it is not clear what the bars and lines refer to in these figures. Moreover, present the results of statistical analysis in the figures. Additionally, it is not clear what the error bars in Figure 6 indicate.

10. The discussion should be more focused as some parts of the discussion extend beyond the studied topic. For instance, the portion on ectomycorrhizal fungus Laccaria laccata (Lines 445–451) discussing its wide host range and phytoprotective capabilities.

11. References cited in the supporting material are not listed; further, list references in line with the journal format. However, presently, the listed references are not uniform as the journal names are presented in full in some instances and abbreviated in others.

Other comments:

12. Line 42: It is better to indicate such large percentage values in folds.

13. Lines 41–44: Combine the sentences.

14. Line 45: Change ‘species’ as ‘fungi’.

15. Line 48: Growth is normally determined in terms of biomass. Therefore, it is not necessary to mention ‘biomass growth increase’.

16. Line 49: Replace ‘growth increase’ with ‘accumulation’ and ‘growth control’ with ‘control’.

17. Line 50: Change ‘showing’ as ‘indicating’.

18. Line 51: Delete ‘also’ and change ‘fungal species’ as ‘fungi’.

19. Line 63–64: Delete the sentence as it is too basic.

20. Line 69: There appears to be something missing. I think it should be ‘......roles as saprophytes and symbionts,........’.

21. Lines 69–73: Combine these into a single sentence.

22. Line 167: Explain the abbreviations at their first mention. ERM- ericoid mycorrhizal fungi.

23. Line 184: What are the ‘liquid isolates’? Were these fungal isolates grown in broths?

24. Line 191: What is the sterile tool? Be more specific.

25. Line 202: Change ‘substances’ as ‘solutions’.

26. Lines 215–217: Cite suitable studies to show the composition and concentrations used exist in large-scale manufacturing plants.

27. Line 224: Mention the duration of drying the material at this temperature.

28. Line 302: The figure title is misleading as what was examined in the present study is the growth response of mycorrhizal fungi and the symbiosis. Therefore modify the figure title accordingly.

29. Line 310: Intraspecific growth responses of what? Fungal isolates?

30. Line 344: Change ‘etc’ as ‘etc.,’.

31. Line 417: Correct the spelling for ‘compounds’.

32. Normalize ‘sp.,’ and ‘spp.,’ throughout the text.

6. PLOS authors have the option to publish the peer review history of their article (what does this mean?). If published, this will include your full peer review and any attached files.

Reviewer #1: **Yes: **Manoj Parihar, ICAR-VPKAS, Almora

Reviewer #2: No

---

## [Author Response · Author response to Decision Letter 0]

25 May 2021

Responses to referee comments

Please find our responses and changes to all referee comments below under each question, stated in blue (in attached document). Line numbers for new text refers to the revised manuscript with track changes.

Reviewer #1: 

1. What a bar represents above column in figure 6? Is this standard error or standard deviation. Kindly provide the detail.

The bars represent standard deviations, and we amended the figure legend, line 450-onwards:

‘Fig 6. Biomass production for three mycorrhizal fungal species grown for three weeks in simulated process water. 

Average biomass production ± stdev for controls grown for one week in Basal Norkrans medium (white bars; n=2), and the simulated process water treatment grown for one initial week in Basal Norkrans medium followed by three weeks in amine solution (grey bars; n=3).’ 

2. Hypothesis and objectives of the present study are not mentioned anywhere in the manuscript. Explain them while concluding the introduction section.

We have amended the following hypothesis to the introduction, line 130-onwards:

‘The perspective of fungi catabolizing hazardous chemical compounds and transforming them to biomass remains understudied and challenging since knowledge is scarce. We hypothesized that fungal species from different ecological groups can survive and grow in the presence of recalcitrant compounds found in wastewaters. Therefor the overall aim of the present study was to test the feasibility of growing fungi for the purpose of metabolizing relevant compounds, utilising part of the fungal species’ diversity in Northern European forests and evaluating their growth and survival on N-containing recalcitrant compounds…’

In the discussion we added the following part, line 462-onward:

…..We confirmed our hypothesis, that both saprotrophic and symbiotic fungal species can survive and grow in the presence of recalcitrant compounds found in wastewaters. Although many isolates were partly restricted or inhibited in growth in the presence of the selected substances, most survived. A subset of three mycorrhizal isolates, which were further tested in a simulated process water experiment, produced large biomass despite exposure to harsh conditions similar to those at which the compounds are present in large-scale manufacturing plants. 

In the summarizing last part of the discussion we added, line 576-onwards:

…Although many isolates were partly restricted or inhibited in growth, most survived in the presence of 2-diethylethanolamine, N3-trimethyl(2-oxiranyl)methanaminium chloride and BAM. The observed growth on these compounds is to our knowledge not previously reported and confirmed the hypothesis that both saprotrophic and symbiotic fungal species can survive and grow in their presence.

3. Why author has selected only three species i.e. Rhizoscyphus ericae, Hygrophorus camarophyllus, and Laccaria laccata AT2001038 in simulated process water experiment while some other species are also performing well in screening experiment.

The second experiment with simulated process water was planned as an initial test following the screening study. We choose to focus on ECM fungal species in this experiment since testing this particular ecological group was one purpose of the study, and most similar studies have focused on saprotrophic fungi. We are aware of the limitations of the two experiments. Ideally, we would see that type of experiment repeated with all analytical procedures in place, since confirming concentrations of the test compounds is central for fully understanding the fungal growth response. At the time of the experiment this was not plausible. In addition, more species in general should be tested in this second setup as well. 

4. The composition and final concentrations of the simulated process water, chosen to reflect the conditions at which the amines are present in large-scale manufacturing plants. It would be great if author may provide some reference data in separate column in table 1 to represent the concentration of these N compounds in different manufacturing plants.

We have added new text in Materials and methods, lines 232-onwards, and a new Table 1 with a separate column reporting estimated values from the industrial wastewater. This is to clarify that we tried to simulate the contents in a particular process water at Cytiva, Uppsala, Sweden.

 “The composition and final concentrations of the simulated process water, chosen to reflect conditions at which the nitrogen containing substances are present in a process water at Cytiva, Uppsala, Sweden, are found in Table 1. To simulate these harsher environments, NaCl and Na2SO4 were added to the 2-diethylaminoethanol and N3-trimethyl(2-oxiranyl)methanaminium chloride, and the C:N ratio in the mixed water was approximately 5:1. The approximative concentrations of the salts in the process water were determined by inductive coupled plasma mass spectrometry (ICP-MS) analyses of chlorine and sulphur contents at ALS Scandinavia, Luleå, Sweden. The concentrations of the nitrogen containing compounds were estimated by their known consumptions” 

Table 1. Composition of simulated process water and estimated concentrations 

in a large-scale process water.

Process water composition (%) (w/v)1 (w/v)2 (w/v)3 

2-diethylaminoethanol 2.4 1.9 1.8 

N3-Trimethyl(2-oxiranyl)methanaminium chloride 1.0 0.8 0.7 

NaCl 1.9 1.5 1.4 

Na2SO4 0.8 0.6 0.5 

1 Original solution

2 After 4:1 dilution with Basal Norkrans liquid medium pH was finally adjusted to 4.5 

3 Estimated concentrations in a large-scale process water (Cytiva, Uppsala, Sweden)

5. The results of this study may be useful for the treatments of industrial wastes. To consider these species as successful biological tool for the degradation of various hazardous nitrogenous compound only on the basis of biomass production is not sufficient and appropriate. However, I know this is just a preliminary study and future advancement in analysing the initial and final concentration of N containing compounds in liquid media and various enzymes responsible for the degradation of these compounds such as amine oxidases would provide direct evidence of catalysing ability of these microbes for investigating compounds.

Yes, we fully agree to this statement. This type of study would need to be followed by more in-depth analyses to understand mechanisms and evaluate the full potential of the method. 

6. At some places in manuscript, concentration of various recalcitrant N compounds is provided as g L-1 or g/L, which needs uniformity.

S1 Fig and S3 Fig legends have been corrected accordingly (g L-1).

7. In figure 1 to 4, lowest concentration of BAM is provided in µL while in supplementary excel file (S2 table1) unit of BAM is given in ml. Kindly check and correct it accordingly.

S2 Table was incoherent, and we have changed it to µL throughout the table. 

8. In reference section, some of the journal names are abbreviated while others are explained fully. Kindly check and revise thoroughly.

All references have been corrected and full names replaced with abbreviations, thank you for the comment.

9. Kindly explain all the abbreviation fully when they appear first time such as ERM.

Abbreviations have been checked and amended; we added in ericoid mycorrhizal fungi (ERM), analysis of variance (ANOVA), and inductive coupled plasma mass spectrometry (ICP-MS).

Reviewer #2: 

1. The concentrations of the nitrogenous compounds (1, 10, 20 g/L) used in the screening study is the major concern in the study. It is not clear how concentrations of these compounds are deduced. Was there any trial run performed to fix the minimum and maximum limit for these compounds?

Since the research area is novel, we did not know at start what concentrations that were tolerable by the fungi. The concentrations of 2-diethylaminoethanol and N3-Trimethyl(2-oxiranyl)methanaminium chloride in a process water at Cytiva, Uppsala, Sweden were estimated to 18 and 7 g L-1 respectively. These concentrations are thus covered by the 1-20 g L -1 concentration interval. 

Included new text line 218-onward: 

‘The choice of these concentrations was based on estimated concentrations of 2-diethylamine and N3-trimethyl(2-oxiranyl)methanaminium chloride (18 and 7.0 g L-1 respectively) in a process water at Cytiva, Uppsala, Sweden’…...

2. The second concern of the study is that the concentration of certain compounds in screening was not determined. The quantity of the BAM present in the experimental solutions is not known as some of the undissolved portions were removed from the solution?

Yes, that is correct for BAM. According to literature (Geyer et al., 1981) 2.7 g L-1 BAM should be possible to dissolve but since we could not replicate this, we opted for preparing a saturated solution by dissolving 250 mg BAM in one litre warm (80o C) double distilled water for 3.5 hours. The undissolved material was removed by vacuum filtration. This is clearly stated in Materials and methods.

3. The rationale for selecting mycorrhizal fungi for stimulated water processing is a bit confusing (Lines 187–190) as mycorrhizal fungi in general exhibited more negative responses than the saprophytic fungi (Lines 422–424). Further, there were saprophytic fungi that had a more positive response to biomass accumulation in the different compound concentrations than Laccaria lacata. It is better to consider the average response of a fungal species to all the exposed chemicals at different concentrations for selection rather than the response to the individual concentration of a chemical.

For ECM fungi in general that was true, but the selected ECM species stood out in the group as their growth response was large. In combination with how easy they are to grow in liquid culture (they commonly produce large amounts of biomass in different types of liquid culture, which we considered important to get the second experiment to work) that formed the basis for selecting the species for the second experiment. Finally, since saprotrophic fungi in general are more documented to metabolise different compounds, we also wanted to test ECM fungal species since they have not previously been considered candidate species for this type of function. 

We added information to Materials and methods, lines 198-onwards:

‘For the simulated process water experiment three fungal species (Hygrophorus camarophyllus, Rhizoscyphus ericae and Laccaria laccata AT2001038) were selected based on growth data in the screening experiment, in combination with how readily the isolates grow in liquid culture (see Results), and were subsequently grown in autoclaved 1000 mL Erlenmeyer flasks….’

4. Please indicate how the fungal isolates obtained directly from field sampling for the study were authenticated. Moreover, mention the accession numbers of the isolates used from the fungal culture collection. At present some codes (these are not accession numbers of the validated isolates?) are mentioned for some taxa in the supplementary table, and it is missing for many.

The fruitbodies collected to obtain new isolates for this study (isolates lacking isolate codes in S1 Table) were identified using classical mycology i.e. morphological characteristics and taxonomic literature. We have field mycological expertise in the author group and are confident in the identifications of these species, none of which is ambiguous. Isolates from earlier studies (as specified in S1 Table) have different sets of isolate codes tracing back to the researchers or collections they originate from. Collections belonging to the Department of Forest Mycology and Plant Pathology are included in earlier published studies, and they are also sequenced to confirm identifications (e.g. since not all are isolated from fruitbodies). We choose not to add in only some accession numbers in the present study, and we did not sequence the new isolates. Based on the fruitbody identification this was not deemed necessary. 

5. Lines 221–222: Justify the reason for harvesting controls after one week of growth. As the fungal growth under any conditions varies with time, compare samples from similar time points. How could you compare the growth of one-week-old fungal culture (control) with four-week-old fungal cultures (treatments)?

Included text line 244-onwards: 

‘The choice of the different growth periods was motivated by the wish to study the more long-term survival and growth of the selected species under harsh conditions, and their ability to suffice with 2-diethylaminoethanol and N3-trimethyl(2-oxiranyl)methanaminium chloride as their C and N sources. It was thus not prioritized to investigate whether the fungi grew better in basal Norkrans medium or not’. 

6. The statistical approach used in the study needs rethinking. For instance, in the screening experiment, there were 48 fungal isolates, three experimental compounds, and different concentrations (including control). Though one-way ANOVA could bring out the variation among the fungal isolates for a tested compound, the reactivity of a fungal isolate to the different compounds and their concentrations tested is obscure. In such a condition, a three-way ANOVA instead of one-way ANOVA would be more meaningful as it could bring out the significance in the variation not only between species but also among compounds and their concentrations. Further, perform Post Hoc analysis for variances that are significant and present the results of post hoc analysis in the figures and tables. As the treatments are compared with a single control, Dunnett’s test would be more appropriate.

We have added a GLM testing potential effects of species, compounds and concentrations, please see line 261-onwards:

Materials and methods:

‘Growth of each fungal species in the N-containing compounds’ treatments was calculated as a percentage of the mean value of the respective growth control (S2 Table). For the screening experiment differences in the average biomass was tested using a general linear model (GLM) with species and compounds as fixed factors and concentration as covariate, and including the interaction terms species*compounds and species*concentration. The interaction terms compounds*concentration and species*compounds*concentration were removed from the model since they could not be estimated. Pairwise comparisons between species, compounds and species*compounds were done using Tukey method. Further, average biomass in control treatment between mycorrhizal and saprotrophic fungi, and between types of saprotrophs (white rot, brown rot and generalists) was tested using One-way analysis of variance (ANOVA), in Minitab 18.1 (Minitab Inc., State College, PA, USA).’

and in the Results, under ‘General growth responses to N-containing compounds’, lines xx:

…’For some treatments where the final biomass production was around zero at harvest, the biomass from the first week of growth on basal Norkrans medium decreased when exposed to the selected compounds. The GLM showed that there were significant overall effects of species (F=13.27, P�0.0001) and compound (F=79.12, P�0.0001) on biomass, as well as a significant interaction between species and compound (F=1.50, P=0.003), but no effect of compound concentration. The model explained 72.25% of the biomass variation. Overall fungi grown in 2-diethylaminoethanol (lowest average biomass) and BAM produced significantly less biomass than in both control and N3-Trimethyl(2-oxiranyl)methanaminium chloride (similar average biomass). Pairwise comparisons for the interaction term species and compound are shown in S3 Table. In general, there were……’

Since the GLM could not estimate the interactions term including concentrations, we have not added the pairwise comparisons directly in the figures, since the comparisons does not match up. Instead, we added and additional S3 Table with this information, showing the species*compounds comparisons.

7. The introduction is a bit long. Concise the introduction by removing facts that are too basic and well-known. Further, it is not clear how the detailed information on the different types of fungi (saprophytic and ectomycorrhizal) to do with the present study. Sum up all this information into one or few s sentences.

We have attempted some re-writing of the introduction, but for us it is a bit unclear what of the facts are too basic and well-know, that would depend on the reader we imagine. Hopefully this is along the lines you intended.

8. Some of the results presented do not agree with the data presented in the tables. For example, Line 263 states that all the saprophytic fungi were basidiomycetes. However, Rhizinia undulata examined in the present study (supplementary table 1) is an ascomycetous fungus.

That was a mistake and has now been corrected in the text, line 297:

‘All saprotrophic fungi, with the exception of Rhizinia undulata, were basidiomycetes.’

9. Figures should be self-explanatory. Instead, details in the figures are missing in the legends presented in the results (insertions). For example, in figures (1-4) treatment names in the x-axis are abbreviated, and it is not clear what the bars and lines refer to in these figures. Moreover, present the results of statistical analysis in the figures. Additionally, it is not clear what the error bars in Figure 6 indicate.

We have added the following information to make figure legends complete for Fig 1-4, line 311-onwards:

‘Fig 1. Mycorrhizal fungal growth responses to recalcitrant amine, amide and ammonium containing media. 

Biomass responses to treatments were compared to controls (C, n=2) for mycorrhizal fungal species in a screening experiment including 2-diethylaminoethanol (2-diet), N3-Trimethyl(2-oxiranyl)methanaminium chloride (gly) and 2,6-dichlorobenzamide (BAM) at three different concentrations (n=1 for each treatment). For 2-diethylaminoethanol and N3-Trimethyl(2-oxiranyl)methanaminium chloride concentrations were 1, 10 and 20 g L-1. For BAM 1µL, 1 mL and 2 mL of a saturated solution was added. Bars show biomass (mg), lines show growth as a percentage of the growth controls. Mycorrhizal species showed positive growth responses to all amine treatments exemplified by……’

For Fig 5, line 408-onwards:

‘Fig 5. Overall fungal biomass responses to the three tested N-containing recalcitrant compounds in the screening experiment. 

Principal component analysis (PCA) showing the variation in biomass responses for 48 fungal isolates when grown for three weeks in control treatment with nutrient solution and N-containing compounds’ treatments (2-diethylaminoethanol [2-diet], N3-Trimethyl(2-oxiranyl)methanaminium chloride [gly] and 2,6-dichlorobenzamide [BAM]) at three different concentrations. For 2-diethylaminoethanol and N3-Trimethyl(2-oxiranyl)methanaminium chloride concentrations were 1, 10 and 20 g L-1. For BAM 1µL, 1 mL and 2 mL of a saturated solution was added. Species differences are visualized by (a) a sample plot with the vector length indicating the relative importance of the amine treatments, and (b) a sample plot with species coded according to functional groups. The first three axes together explained 87.5% of the total variation (84181.4).’

For Fig 6, line 450-onwards:

See comments above for a question from referee #1.

For Fig S4, lines 930-onwards:

‘S4 Fig. Principal component analysis (PCA) showing the overall variation in fungal biomass responses to three recalcitrant N-containing compounds. 

48 fungal isolates when grown for three weeks in control treatment with nutrient solution and amine treatments (2-diethylaminoethanol [2-diet], N3-Trimethyl(2-oxiranyl)methanaminium chloride [gly] and 2,6-dichlorobenzamide [BAM]) at three different concentrations. For 2-diethylaminoethanol and N3-Trimethyl(2-oxiranyl)methanaminium chloride concentrations were 1, 10 and 20 g L-1. For BAM 1µL, 1 mL and 2 mL of a saturated solution was added. Species differences are visualized by a sample plot with circle size depicting the average biomass response across all treatments. The first three axes together explained 87.5% of the total variation (84181.4).’

10. The discussion should be more focused as some parts of the discussion extend beyond the studied topic. For instance, the portion on ectomycorrhizal fungus Laccaria laccata (Lines 445–451) discussing its wide host range and phytoprotective capabilities.

The text section about Laccaria laccata has been shortened down to focus the discussion, line 509-onwards:

‘Laccaria laccata, one of the ECM species included in the simulated process water experiment, is known to be easy to grow in liquid culture [55] and has shown potential as biological control agent against disease causing fungi [57]. Although ECM fungi such as Laccaria laccata degrade pollutants and expedite removal of persistent organic pollutants [58, 59], it is unknown whether the species can metabolise amines, amides or quaternary ammonium compounds….’

11. References cited in the supporting material are not listed; further, list references in line with the journal format. However, presently, the listed references are not uniform as the journal names are presented in full in some instances and abbreviated in others.

The reference list has been updated accordingly, please see comments under the responses to referee #1. References in supporting material are already listed: for S1 Table we have added the numbers for the reference list to make consistent with the manuscript. 

Other comments:

12. Line 42: It is better to indicate such large percentage values in folds.

Changed to:

‘a 35-fold and 4.5-fold increase in biomass, respective…’

13. Lines 41–44: Combine the sentences.

Changed to:

The ericoid (ERM) mycorrhizal fungus Rhizoscyphus ericae showed the best overall growth on 2-diethylaminoethanol and BAM in the 1-20 g L-1 concentration range, with a 35-fold and 4.5-fold increase in biomass, respectively.

14. Line 45: Change ‘species’ as ‘fungi’.

Changed to:

‘…fungi…’

15. Line 48: Growth is normally determined in terms of biomass. Therefore, it is not necessary to mention ‘biomass growth increase’.

Changed to:

‘…biomass increase….’

16. Line 49: Replace ‘growth increase’ with ‘accumulation’ and ‘growth control’ with ‘control’.

Done.

17. Line 50: Change ‘showing’ as ‘indicating’.

Done.

18. Line 51: Delete ‘also’ and change ‘fungal species’ as ‘fungi’.

Done.

19. Line 63–64: Delete the sentence as it is too basic.

Done.

20. Line 69: There appears to be something missing. I think it should be ‘......roles as saprophytes and symbionts,........’.

Changed to;

‘Fungi are of fundamental importance to all ecosystems in terms of elemental cycling, and evolution of primary lifestyles (saprotrophic and symbiotic fungi) has occurred repeatedly via loss or reduction of genes for groups of enzymes [3, 4]. Saprotrophic fungi primarily facilitate organic matter decomposition, utilizing carbon (C) and….’

21. Lines 69–73: Combine these into a single sentence.

We changed the text to (see above):

‘Fungi are of fundamental importance to all ecosystems in terms of elemental cycling, and evolution of primary lifestyles (saprotrophic and symbiotic fungi) has occurred repeatedly via loss or reduction of genes for groups of enzymes [3, 4]. Saprotrophic fungi primarily facilitate organic matter decomposition, utilizing carbon (C) and….’

22. Line 167: Explain the abbreviations at their first mention. ERM- ericoid mycorrhizal fungi.

Done.

23. Line 184: What are the ‘liquid isolates’? Were these fungal isolates grown in broths?

Changed to:

‘…liquid cultures…’, the Basal Norkrans media does not contain any agar and is hence liquid instead of solid.

24. Line 191: What is the sterile tool? Be more specific.

Changed to:

‘…scalpel…’

25. Line 202: Change ‘substances’ as ‘solutions’.

Done.

26. Lines 215–217: Cite suitable studies to show the composition and concentrations used exist in large-scale manufacturing plants.

Since the application area is novel, both regarding selected species and methodology, there are no similar studies that to the best of our knowledge are relevant. However, in the introduction section, there are several studies including various application areas that are cited. Furthermore, the choice of nitrogen containing compounds in the present study is unique, since it is connected with the large-scale production of protein separation media at Cytiva, Uppsala, Sweden. 

27. Line 224: Mention the duration of drying the material at this temperature.

Added:

‘…for 24 hrs….’

28. Line 302: The figure title is misleading as what was examined in the present study is the growth response of mycorrhizal fungi and the symbiosis. Therefore modify the figure title accordingly.

Done:

‘….mycorrhizal fungal growth…’

29. Line 310: Intraspecific growth responses of what? Fungal isolates?

Changed to:

‘Fig 4. Intra-specific saprotrophic fungal growth responses…’

30. Line 344: Change ‘etc’ as ‘etc.,’.

Done.

31. Line 417: Correct the spelling for ‘compounds’.

Done.

32. Normalize ‘sp.,’ and ‘spp.,’ throughout the text.

Done.

---

## [Decision Letter · Decision Letter 1]

28 Jun 2021

PONE-D-20-39694R1

Survival and growth of saprotrophic and mycorrhizal fungi in recalcitrant amine, amide and ammonium containing media

PLOS ONE

Dear Dr. Fransson,

Thank you for submitting your manuscript to PLOS ONE. After careful consideration, we feel that it has merit but does not fully meet PLOS ONE’s publication criteria as it currently stands. Therefore, we invite you to submit a revised version of the manuscript that addresses the points raised during the review process.

I have completed the editorial review of your manuscript, and a summary is appended below. The reviewers recommend reconsideration of your paper following minor revision. 

Please pay particular attention to the comment regarding the needful corrections. Failure to do so will result in the delay in the further review of your manuscript. 

We look forward to receiving your revised manuscript.

Kind regards,

Vijai Gupta, PhD in Microbiology

Academic Editor

PLOS ONE

Journal Requirements:

Additional Editor Comments (if provided):

I have completed the editorial review of your manuscript, and a summary is appended below. The reviewers recommend reconsideration of your paper following minor revision.

Please pay particular attention to the comment regarding the needful corrections. Failure to do so will result in the delay in the further review of your manuscript.

Reviewers' comments:

Reviewer's Responses to Questions

**Comments to the Author**

1. If the authors have adequately addressed your comments raised in a previous round of review and you feel that this manuscript is now acceptable for publication, you may indicate that here to bypass the “Comments to the Author” section, enter your conflict of interest statement in the “Confidential to Editor” section, and submit your "Accept" recommendation.

Reviewer #1: (No Response)

Reviewer #2: All comments have been addressed

2. Is the manuscript technically sound, and do the data support the conclusions?

Reviewer #1: Partly

Reviewer #2: Yes

3. Has the statistical analysis been performed appropriately and rigorously? 

Reviewer #1: Yes

Reviewer #2: Yes

4. Have the authors made all data underlying the findings in their manuscript fully available?

Reviewer #1: Yes

Reviewer #2: Yes

5. Is the manuscript presented in an intelligible fashion and written in standard English?

Reviewer #1: Yes

Reviewer #2: Yes

6. Review Comments to the Author

Reviewer #1: (No Response)

Reviewer #2: In this revised version, the authors have taken into consideration all the suggestions raised in my previous review and modified the manuscript accordingly. Nevertheless, there are a few minor changes necessary as mentioned below.

Line 74: Change ‘Ectomycorrhizal fungi’ to ‘Ectomycorrhizal fungi (ECM),’.

Lines 201, 226: Use the ‘degree symbol’.

Line 211: Subscript the numbers in the chemical formula.

Lines 129, 199, 288, 305, 322, 333-334, 386: It is not clear why the abbreviation for 2,6-dichlorobenzamide is introduced so many times as it is already there in Line 31.

Lines 347, 359–362: Express these huge percentage values in folds.

Line 372: Change ‘that is appears’ as ‘that it appears’.

Line 433: Replace ‘[5, 8]’ with ‘[5, 8],’.

Line 434: Correct the spelling for ‘to’.

7. PLOS authors have the option to publish the peer review history of their article (what does this mean?). If published, this will include your full peer review and any attached files.

Reviewer #1: **Yes: **Manoj Parihar

Reviewer #2: No

---

## [Author Response · Author response to Decision Letter 1]

2 Aug 2021

Response to reveiwers

Please see below the final changes made to the manuscript. Our responses are stated in blue.

Reviewer #2: In this revised version, the authors have taken into consideration all the suggestions raised in my previous review and modified the manuscript accordingly. Nevertheless, there are a few minor changes necessary as mentioned below.

Line 74: Change ‘Ectomycorrhizal fungi’ to ‘Ectomycorrhizal fungi (ECM),’.

Done.

Lines 201, 226: Use the ‘degree symbol’.

Done.

Line 211: Subscript the numbers in the chemical formula.

Done.

Lines 129, 199, 288, 305, 322, 333-334, 386: It is not clear why the abbreviation for 2,6-dichlorobenzamide is introduced so many times as it is already there in Line 31.

Yes, thank you for pointing that out. We kept the introduction of compound and abbreviation in the Introduction, line 129 which is the first place it is mentioned in the main body of the text. All other places have been replaced with BAM.

Lines 347, 359–362: Express these huge percentage values in folds.

Done.

Line 372: Change ‘that is appears’ as ‘that it appears’.

Done.

Line 433: Replace ‘[5, 8]’ with ‘[5, 8],’.

Done.

Line 434: Correct the spelling for ‘to’.

Done.

Attached letter: Although authors has improved the manuscript markedly but still I have some queries and doubt regarding my earlier comments. 

What is the history of sampling site? I mean fungal species isolated from any waste water disposal site would be more adaptable for degrading the recalcitrant compound compared to fungi isolated from healthy site.

Fungal collection was made in forests and other natural habitats for the sake of isolating the fungal species in their natural environment. We had an ecological angle into this study. If our aim had been to target polluted sites this would have been done differently, yes. 

Introduction needs to be concise. 

The Introduction was revised the first round, shortened, and focused according to suggestions. We are not sure what part of the Introduction you consider unconcise. Therefore, it is difficult to make more changes.

Your earlier reply regarding selection of three species i.e. Rhizoscyphus ericae, Hygrophorus camarophyllus, and Laccaria laccata AT2001038 is bit confusing. As you mentioned that your focus was on ECM fungal species because lots of work has been already done on saprophytic fungi. If this is true then why you have screened a large number of saprophytic fungi in your preliminary study. 

The way into this study was dual: firstly, we have a purely ecological focus in that we wanted to screen relatively broadly across a number of fungal species that are commonly found in European forests. In this way we produce data that can be put into both an ecological and taxonomical context, since knowledge is limited. Although lots of work has been done on saprotrophic fungi, this work has mostly focused on very few saprotrophic species. Secondly, we wanted to focus partly on mycorrhizal fungi since they are not normally included in this type of studies, despite having similar enzyme systems as the saprotrophic fungi. 

In table 1, 2, 6-dichlorobenzamide (BAM) composition is not mentioned anywhere. During BAM preparation you have dissolved 250 mg BAM in 1 liter water but later undissolved material was removed by filtration. This would affect concentration of BAM in prepared solution and by doing so final concentration of BAM cannot be figure out. 

BAM was not included in the simulated process water experiment, hence it is not mentioned in Table 1.

Figure 1 and 2 are still confusing. Simply we can’t predict what bar represent means biomass or growth. Similarly, on X axis unit is given for BAM but not for other two compounds. It must be symmetrical. 

In the figure legend it is already clearly stated that:

‘…Bars show biomass (mg), lines show growth as a percentage of the growth controls.’

This should be enough to explain the figure. We corrected the figures and removed the units for BAM. The information is now only included in the figure legend. 

Some of the sentences needs to be corrected such as:

Line: 313 to 315

Changed to: Negative growth responses to all treatments were exemplified by (e) Ganoderma applanatum (brown rot) and (f) Lenzites betula (white rot), as well as two parasites (g) Phanerochaete sordida, and (h) Rhizina undulata.

Line: 434

In this study, we are predicting fungal ability to degrade the recalcitrant N compound just by their growth and biomass. However, we have to assess the final concentration of N compound in growth medium after completion of the experiment. Then only we can say with certainty that how much our fungal species are effective in degrading the studied compound. 

We assume you wanted something along those lines appended to the discussion? We already have information about this issue towards the end of the Discussion, and in the final summery paragraph. We now added the following to the end of the starting Discussion paragraph, line 440 onwards:

‘…. In this study, we are predicting fungal ability to degrade the recalcitrant N compound by their growth and biomass. However, this first screening study needs to be followed by more in-depth studies confirming decreased concentrations of these substances.’

Species identification by morphological evidences is good but for certainty and better reliability it must be supported by molecular sequencing. 

Here we need to agree to disagree, application of molecular tools is not always the most reliable way to identify fungal species. As an example the NCBI database is commonly used for identifying environmental sequences from soil amplicons, field samples etc. Unfortunately, there are many wrongly named sequences in that database (even things that are not truly fungal are stated as fungal), and one would need to use more curated information in e.g. UNITE (where entries have been confirmed by mycologists specialising in different parts of fungal taxonomy, then produced sequences from fungal fruitbody material which is correctly labelled from the beginning). When working with environmental sequences they can still be grouped into species hypothesis etc and compared top other thus grouped sequences, but we do believe that the faith in molecular tools over classical mycology is a bit inflated. In the case of the species included in the present study there are no ambiguous species that we were not sure of. The persons responsible for collecting, identifying and isolating the species are highly competent in the area of classical mycology as well as molecular ecology/fungal community ecology by molecular methods. We are confident that the identifications were done correctly. Finally, the study was conducted some years ago and we do not intend sequence the isolates at this point. We hope this is acceptable.

---

## [Editor Report · Decision Letter 2]

16 Aug 2021

Survival and growth of saprotrophic and mycorrhizal fungi in recalcitrant amine, amide and ammonium containing media

PONE-D-20-39694R2

Dear Dr. Fransson,

We’re pleased to inform you that your manuscript has been judged scientifically suitable for publication and will be formally accepted for publication once it meets all outstanding technical requirements.

Kind regards,

Vijai Gupta, PhD in Microbiology

Academic Editor

PLOS ONE

Additional Editor Comments (optional):

Authors have answered most of the important reviewer's queries as raised against the original version. The present version is well structured.
---

## [Editor Report · Acceptance letter]

24 Aug 2021

PONE-D-20-39694R2 

Survival and growth of saprotrophic and mycorrhizal fungi in recalcitrant amine, amide and ammonium containing media 

Dear Dr. Fransson:

I'm pleased to inform you that your manuscript has been deemed suitable for publication in PLOS ONE. Congratulations! Your manuscript is now with our production department. 

Kind regards, 

on behalf of

Dr. Vijai Gupta 

Academic Editor

PLOS ONE